# MARS: Modular Agent with Reflective Search for Automated AI Research

Jiefeng Chen [1]   Bhavana Dalvi Mishra [1]   Jaehyun Nam [1]   Rui Meng [1]   Tomas Pfister [1]   Jinsung Yoon [1]

## Abstract

A critical bottleneck in automating AI research is the execution of complex machine learning engineering (MLE) tasks. MLE differs from general software engineering due to computationally expensive evaluation (e.g., model training) and opaque performance attribution. Current LLM-based agents struggle here, often generating monolithic scripts that ignore execution costs and causal factors. We introduce **MARS** (**M**odular **A**gent with **R**eflective **S**earch), a framework optimized for autonomous AI research. MARS relies on three pillars: (1) Budget-Aware Planning via cost-constrained Monte Carlo Tree Search (MCTS) to explicitly balance performance with execution expense; (2) Modular Construction, employing a "Design-Decompose-Implement" pipeline to manage complex research repositories; and (3) Comparative Reflective Memory, which addresses credit assignment by analyzing solution differences to distill high-signal insights. MARS achieves state-of-the-art performance among open-source frameworks on MLE-Bench under comparable settings, maintaining competitiveness with the global leaderboard's top methods. Furthermore, the system exhibits qualitative "Aha!" moments, where 63% of all utilized lessons originate from cross-branch transfer, demonstrating that the agent effectively generalizes insights across search paths.

## 1. Introduction

The integration of Large Language Models (LLMs) into software engineering has fundamentally transformed code generation, evolving from simple auto-completion to autonomous agents capable of resolving GitHub issues (Jimenez et al., 2023; Yang et al., 2024) and generating functional scripts (Li et al., 2022; Wang et al., 2025; Jiang et al., 2025). However, while current agents excel at general software maintenance tasks – such as patching bugs or writing unit tests – they face significant hurdles when applied to the domain of Automating AI Research (Chan et al., 2025; Tian et al., 2024; Wijk et al., 2025; Yamada et al., 2025; Starace et al., 2025). A critical bottleneck in this domain is the execution of complex machine learning engineering (MLE) tasks. Unlike standard software development, where correctness is often binary and verification is computationally cheap, MLE is a probabilistic, resource-intensive endeavor. It requires not only coding intelligence but also the strategic foresight to navigate a landscape defined by computationally expensive evaluations, opaque performance attribution, and high architectural complexity.

Existing agentic frameworks, designed primarily for monolithic code generation, struggle to adapt to these constraints. First, they typically view problem-solving as a purely code-based challenge (Huang et al., 2024; Jiang et al., 2025; Toledo et al., 2026), ignoring the economic reality of research: model training and data processing consume vast computational resources. An agent that improves model accuracy by 0.1% but increases training time from one hour to ten hours is often practically useless, yet standard search algorithms would prioritize it. Second, the monolithic and unstructured scripts often produced by previous LLM agents are fragile and ill-suited for the modular complexity required in research repositories, where data loading, model architecture, and training loops must interact seamlessly. Finally, research progress is iterative and opaque; when a new experiment yields better results, it is difficult to isolate the causal factor. Standard memory-based agents (Packer et al., 2023; Shinn et al., 2023; Xu et al., 2026; Ouyang et al., 2026) lack the mechanism to solve this *credit assignment problem*, often failing to learn effectively from past trials.

To bridge this gap, we introduce **MARS** (**M**odular **A**gent with **R**eflective **S**earch), a framework explicitly optimized for the distinct constraints of autonomous scientific discovery. MARS reformulates the research process as a search for an optimal software repository, governed by three core pillars. To address the high cost of evaluation, we employ *Budget-Aware Planning* via a cost-constrained Monte Carlo Tree Search (MCTS). Unlike general search algorithms, our method explicitly balances performance maximization with

---

[1]Google Cloud AI Research. Correspondence to: Jiefeng Chen <jiefengc@google.com>, Jinsung Yoon <jinsungyoon@google.com>.

*Proceedings of the 43$^{rd}$ International Conference on Machine Learning*, Seoul, South Korea. PMLR 306, 2026. Copyright 2026 by the author(s).

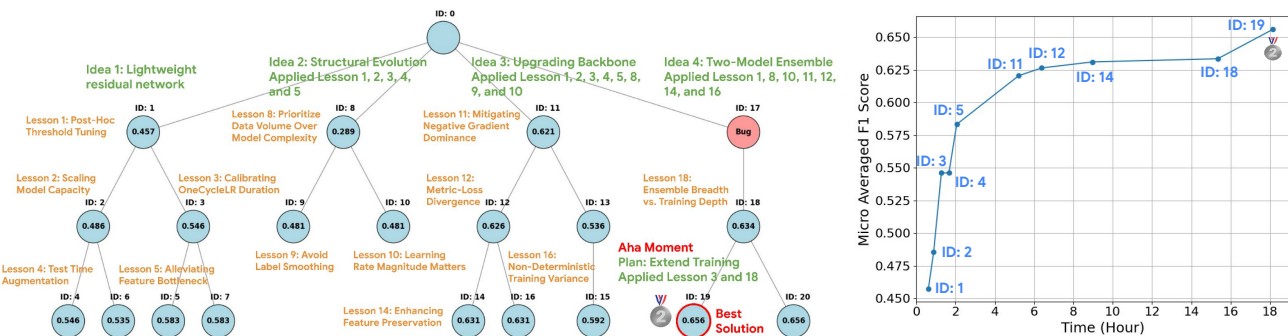

*Figure 1.* The "Aha!" moment of MARS on the challenging iMet-2020-FGVC7 task. The visualization tracks validation performance gains triggered by specific strategic lessons. While existing methods fail to reach medal-level performance, MARS progressively refines its strategy – evolving from a lightweight residual network to model ensemble techniques – to ultimately achieve a silver medal.

execution expense, prioritizing efficient solutions – such as favoring a 1-hour training run over a 4-hour run if performance is comparable – to optimize the discovery rate within a fixed budget. To manage architectural complexity, we replace fragile scripting with a *Modular "Design-Decompose-Implement"* pipeline. This structure employs specialized agents to architect solutions into independent, testable modules. Finally, to resolve the credit assignment problem, we introduce *Comparative Reflective Memory*. By analyzing the differences between the current solution and the best-known solution, the agent distills high-signal, causal insights, isolating the specific factors driving performance shifts in a way that standard memory mechanisms cannot. As illustrated in Figure 1, these pillars allow MARS to experience "Aha!" moments during long-horizon exploration, successfully navigating complex optimization landscapes where baselines fail.

Our contributions are summarized as follows:

- We introduce **MARS**, a framework aimed at automating AI research by optimizing for complex MLE tasks, featuring a novel combination of Budget-Aware MCTS, a modular implementation pipeline, and Comparative Reflective Memory.

- We perform extensive evaluation on the MLE-Bench benchmark, where MARS achieves state-of-the-art performance among open-source frameworks under comparable settings. Ablation studies further validate the necessity of each proposed mechanism.

- We provide qualitative analyses of how MARS drives long-horizon exploration. To facilitate future research, we release prompts in Appendix H, and MARS generated code, trajectories in https://github.com/jfc43/MARS.

## 2. Related Work

**Automated AI Research & MLE Bottleneck.** Recent advancements in LLMs have driven ambitious efforts to automate complex, long-horizon AI research (Lu et al., 2024; Yamada et al., 2025; Weng et al., 2025; Tang et al., 2026; Schmidgall et al., 2025; Pu et al., 2025; Starace et al., 2025). Central to these efforts is the realization that Machine Learning Engineering (MLE) serves as a critical execution bottleneck (Chan et al., 2025; Wijk et al., 2025); without robust autonomous MLE capabilities, higher-level research automation cannot succeed. While numerous agentic frameworks have been proposed to address these challenges (Jiang et al., 2025; Toledo et al., 2026; Yang et al., 2025; Liu et al., 2025b; Team et al., 2025; Li et al., 2025; Nam et al., 2026; Zhu et al., 2026), existing systems predominantly operate under a *monolithic paradigm*, generating expansive, single-file scripts. This approach typically results in fragile codebases that lack the modularity essential for rigorous engineering. MARS departs from this by enforcing a *repository-level paradigm* that systematically decomposes tasks into distinct, testable, and maintainable modules, mirroring the professional software architecture required for scalable AI research.

**Search Algorithms in Code Generation.** Solving long-horizon AI research problems, where code execution is resource-intensive, necessitates effective search strategies for code optimization. While various algorithms have been adapted for these systems – including greedy search (Jiang et al., 2025), Monte Carlo Tree Search (MCTS) (Kocsis & Szepesvári, 2006; Liu et al., 2025b), and Evolutionary search (Team et al., 2025) – they typically optimize solely for task performance, neglecting computational cost. While recent work has introduced "budget awareness" for tool-augmented agents via external plug-ins (Liu et al., 2025a), such methods are primarily designed for discrete actions like web search. In contrast, we introduce *Budget-aware MCTS*, which integrates an efficiency-guided reward function di-

rectly into the search tree. This allows MARS to balance the exploitation of high-performing strategies with the exploration of novel ideas, penalizing computationally expensive solutions to ensure both performance and efficiency.

**Reflective Learning and Memory.** Enabling agents to improve iteratively through environmental interaction is a rapidly evolving research area. Early approaches, such as Reflexion (Shinn et al., 2023), focus on self-correction using verbal reinforcement derived from prior mistakes. Recently, the field has expanded to include sophisticated memory and distillation frameworks. For instance, several works explore advanced memory architectures for personalization and associative retrieval (Tan et al., 2025; Xu et al., 2026), while others focus on refining process memory and distilling execution traces into stable strategies (Forouzandeh et al., 2025; Ouyang et al., 2026; Zhu et al., 2026). Closer to our domain, Zhao et al. identify "good practices" by comparing high-level action trajectories, and Jansen et al. cache useful code blocks for future reuse. MARS advances this paradigm by introducing "Lesson Learning". Unlike prior methods that summarize execution logs to debug errors or track binary success-failure trajectories, our method explicitly analyzes the causal link between *code changes* and performance variations. This comparative analysis isolates effective algorithmic changes from confounding factors, distilling high-value insights into a lesson pool to guide future exploration.

Table 1 summarizes the key differences between MARS and existing MLE agent frameworks.

## 3. Problem

We first formalize the general problem of *Long-Horizon Agentic Problem Solving*, where an autonomous agent is tasked with constructing a complex artifacts (e.g., a software system) to satisfy a set of requirements within a constrained budget. Let $\mathcal{P}$ denote a problem instance defined by the tuple $\mathcal{P} = (\mathcal{I}, \mathcal{E}, \mathcal{O})$, where: (1) $\mathcal{I}$ represents the *Instruction* or requirements provided in natural language. (2) $\mathcal{E}$ denotes the *Environment* with which the agent interacts to validate its solutions. This can be a compiler, a simulator, or a dataset depending on the task scenario. (3) $\mathcal{O}$ is the *Objective* function that quantifies the quality of the solution.

The goal is to find a solution $s^*$ that maximizes $\mathcal{O}$ by interacting with $\mathcal{E}$, subject to a cost constraint $B$ (e.g., time budget or monetary cost):

$$s^* = \arg\max_{s} \quad \mathcal{O}(s, \mathcal{E}), \quad s.t. \quad \text{Cost}(s) \leq B \quad (1)$$

where the search space for $s$ is often vast and unstructured (e.g., the space of all possible Python programs).

**MLE Task Scenario.** Machine Learning Engineering (MLE) is a representative and challenging instantiation of this general problem class. MLE requires the agent to engineer a full pipeline that processes data, trains models, and validates results. In this scenario, $\mathcal{E}$ consists of the provided datasets while $\mathcal{O}$ is the performance metric (e.g. accuracy) on the held-out test set. Refer to Appendix B for details.

## 4. Method

### 4.1. Overall Framework

We propose MARS, a general agent scaffolding framework designed to enable autonomous agents to solve long-horizon AI Research problems, as illustrated in Figure 2. Formally, we define the problem as a tuple $\mathcal{P} = (\mathcal{I}, \mathcal{E}, \mathcal{O})$, where the agent must follow the instruction $\mathcal{I}$ within an environment $\mathcal{E}$ to maximize an objective $\mathcal{O}$ under a cost budget $B$. To address the core challenges of exploration complexity, context management, and solution robustness in this setting, our framework integrates three key capabilities:

- **Modular Construction Strategy:** Instead of generating monolithic scripts, we enforce a structured, repository-level software architecture. This paradigm allows for handing complex logic with greater accuracy, efficient code reuse, and improving testability.

- **Comparative Reflective Memory:** To overcome context window limitations, we introduce a "Lesson Learning" mechanism that distills high-value, causal insights from past interactions (both successes and failures) into a compact, retrievable knowledge base.

- **Resource-Aware Planning:** We employ a budget-aware Monte Carlo Tree Search (MCTS) algorithm to systematically explore the solution space. This allows the system to balance the exploitation of promising candidates with the exploration of novel ideas, preventing local optima, and penalize solutions that are costly.

### 4.2. Modular Decomposition

A primary contribution of this work is the strategic shift from generating monolithic scripts to a Modular Implementation paradigm. This paradigm addresses several inherent limitations of LLM-based coding. First, it bypasses token output limits by distributing code across multiple files. Second, it enhances precision; by focusing on smaller logical units, the agent encounters less context noise and can handle complex logic with greater accuracy. Third, it enables efficiency via caching, as validated modules can be reused without regeneration. Finally, it significantly improves testability, as debugging is localized to specific files rather than requiring full-script diagnosis.

We define a node solution $s_n$ as a tuple comprising a set of

| System | Modular? | Budget-Aware Search? | Memory Mechanism |
|---|---|---|---|
| AIDE (Jiang et al., 2025) | ✗ | ✗ | All previous designs, scores, and notes |
| MLE-STAR (Nam et al., 2026) | ✗ | ✗ | Some previous plans, code and results |
| AIRA (Toledo et al., 2026) | ✗ | ✗ | Scoped Memory: some previous designs, scores, and notes |
| R&D-Agent (Yang et al., 2025) | ✗ | ✗ | Collaborative Memory: previous solutions, results, and insights |
| ML-Master 2.0 (Zhu et al., 2026) | ✗ | ✗ | Hierarchical Cognitive Caching: scripts, facts, strategies |
| **MARS (Ours)** | ✓ | ✓ | **Comparative Reflective Memory: solution & debug lessons** |

*Table 1.* Comparison of MLE agents in terms of: 1) Do they generate modular code? 2) Do the agents take into account runtime/budget during search? 3) What types of memory mechanisms do they use to enhance performance on a given task? (✓: yes, ✗: no).

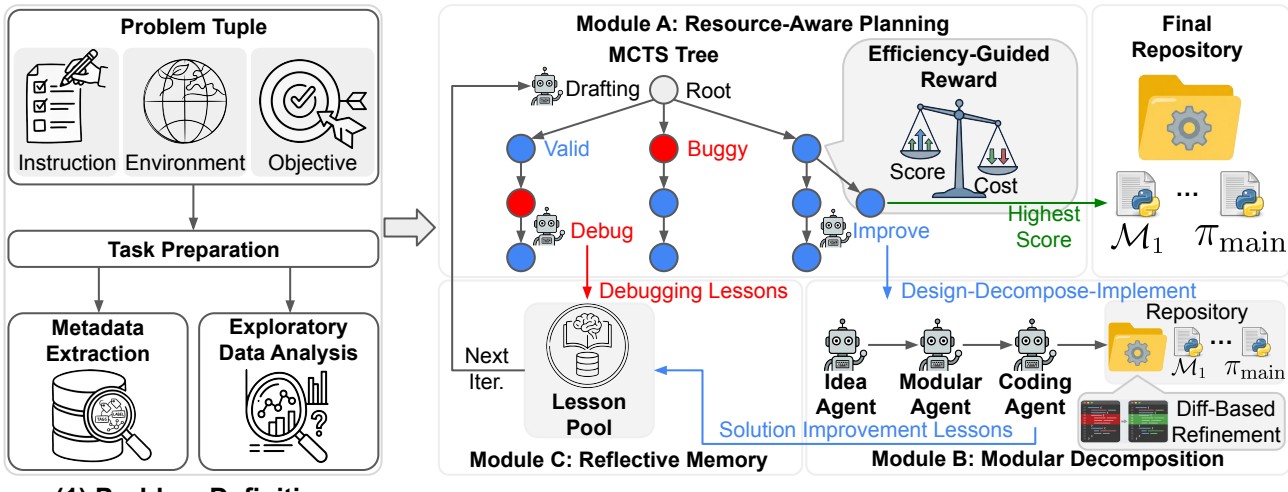

**(1) Problem Definition & Preparation**

**(2) MARS Framework Loop**

*Figure 2.* Overview of the MARS Framework. MARS reformulates long-horizon coding as a search for an optimal software repository. (1) Task Preparation: The agent grounds the abstract problem (Instruction, Environment, Objective) tuple by exploratory analysis of the given dataset and metadata. (2) The MARS Loop: The agent iteratively evolves solutions through three synergistic modules: **(A) Resource-Aware Planning:** A Budget-Aware MCTS strategically navigates the search space by selecting actions from {Draft new architecture, Debug runtime errors, Improve a valid solution}. It optimizes an efficiency-guided reward that explicitly balances performance maximization with the penalty of high execution costs. **(B) Modular Decomposition:** To replace fragile monolithic scripting, the system employs a "Design-Decompose-Implement" pipeline. Specialized {Idea, Modular, Coding} agents architect the solution into independent, testable modules. This structure enables precise Diff-Based Refinement, allowing the agent to update specific logic blocks without regenerating the entire codebase. **(C) Reflective Memory:** This module distills raw execution logs into structured Debugging and Solution Lessons to proactively prevent error repetition and accelerate convergence in later iterations.

$l$ independent modules and one orchestration script:

$$s_n = \langle \{\mathcal{M}_j\}_{j=1}^l, \pi_{main} \rangle \qquad (2)$$

Each module $\mathcal{M}_j$ encapsulates a specific sub-task (e.g., data preprocessing, configuration), while the main script $\pi_{\text{main}}$ orchestrates the end-to-end pipeline.

To instantiate this structure, we employ a three-stage "Design-Decompose-Implement" workflow:

- **Idea Generation:** An *Idea Generation Agent* articulates a comprehensive natural language plan covering various aspects of the solution.

- **Module Decomposition:** A *Modular Agent* parses the plan and decomposes the solution into logical, independent functional modules.

- **Component Implementation and Debugging:** A *Coding Agent* sequentially implements each module $\mathcal{M}_j$, employing a validation script to debug and verify functionality. Once validated, the agent orchestrates the modules via the main script $\pi_{\text{main}}$.

To prevent wasteful full-repository regeneration, we adopt a Diff-Based Editing mechanism. Code modifications are structured in a standardized diff format, specifying the target file, the block to replace, and the new code. This enables atomic, multi-file updates in a single inference step.

### 4.3. Comparative Reflective Memory

Solving complex tasks requires long-horizon exploration, generating extensive interaction trajectories that often exceed context window constraints. More importantly, research progress is inherently iterative and opaque; when a

new experiment yields improved results, isolating the specific causal factors remains a challenge. Standard memory-based agents often lack the mechanisms to solve this credit assignment problem, failing to learn effectively from past trials. To address this, we propose *Comparative Reflective Memory*, a mechanism designed to distill high-signal, causal insights from the exploration process into a compact lesson pool.

**Solution Improvement via Comparative Reflection.** We employ a two-stage process to resolve the credit assignment problem by synthesizing lessons from valid solutions. First, an *Empirical Analysis Agent* reviews execution logs to extract objective findings (e.g., metric trends). Subsequently, a *Lesson Distillation Agent* performs a comparative reflection by analyzing the delta between the current solution and the previous best-known solution. This isolates the specific algorithmic changes driving performance shifts, resulting in a structured lesson containing: (1) The isolated causal change, (2) A comparative impact analysis, and (3) A generalized rule for future iterations.

**Debugging Lessons.** For failed executions, a dedicated agent analyzes the buggy code, error logs, and the applied fix. It outputs a lesson confirming the fix's efficacy, explaining the failure logic, and providing guidelines to preemptively identify similar errors.

**Lesson Management.** To maintain a high-signal lesson pool, a Review Agent evaluates new lessons against the existing pool through LLM-based reasoning, filtering out redundant insights to ensure the retrieved context remains diverse and relevant.

**Lesson Utilization.** When executing solution improvement or debugging actions, the agent utilizes relevant knowledge from the corresponding lesson categories. We retain the $K_m$ most recent lessons in the agent's memory to manage context. To ensure interpretability, the agent is instructed to explicitly cite specific lessons whenever they are applied.

## 4.4. Budget-Aware MCTS

We adopt the Monte Carlo Tree Search (MCTS) framework to explore the solution space, which iterates through four phases: Selection, Expansion, Simulation, and Backpropagation. In this section, we detail our domain-specific modifications: (1) specialized expansion operators, (2) a coherent node selection strategy, and (3) an *Efficiency-Guided Reward Function* that balances performance with cost. Appendix C provides a review of standard MCTS principles.

### 4.4.1. ACTIONS AND EXPANSION

We define three distinct operators to transform a parent state $s_{parent}$ into a child solution $s_{new}$:

- **Drafting (Root Expansion):** Generates a completely new solution $s_{new}$ from scratch.

- **Improvement:** Applied to valid, executable nodes. The agent modifies the modules and the main script from $s_{parent}$ to maximize the objective $\mathcal{O}$.

- **Debugging:** Applied to nodes where execution failed. The agent inherits the solution structure from $s_{parent}$ but modifies specific modules or the orchestration script to resolve runtime errors. Buggy children enter an automatic debugging loop with up to $N_d$ debugging actions to fix the errors.

### 4.4.2. NODE SELECTION

We employ the Upper Confidence Bound for Trees (UCT) algorithm to navigate the solution space, balancing the exploitation of high-performing solutions with the exploration of new solutions.

The selection phase begins at the root node. In each step, we select the child node that maximizes the UCT value. This traversal continues recursively until we identify a candidate node, defined as a node that is not yet "fully expanded".

The root node is set fully expanded unless any of the follow condition occurs: (1) It does not have any children; (2) the best solution has not been improved after implementing $n_s$ valid nodes.

If the traversal reaches a leaf node that is already fully expanded (it implies that no further debugging or improvement is permitted for that branch), then the root node is re-activated to allow for new drafts.

The buggy nodes are always set fully expanded. The valid nodes are set fully expanded if they have $\geq N_i$ children (attempts to improve).

### 4.4.3. EFFICIENCY-GUIDED REWARD FUNCTION

To guide the search efficiently, we design a reward function $R(v)$ that rewards performance gains and penalizes long execution time. Let $M(v)$ denote the performance metric of a node $v$, and let $t(v)$ and $L(v)$ represent its execution time and time limit, respectively. We first normalize the performance metric relative to the history of explored nodes $\mathcal{V}$. Let $M_{max} = \max_{v' \in \mathcal{V}} M(v')$ and $M_{min} = \min_{v' \in \mathcal{V}} M(v')$. We define the global normalized score $G(v)$ as:

$$G(v) := \begin{cases} 0.5 & \text{if } M_{max} = M_{min}, \\ \frac{M(v) - M_{min}}{M_{max} - M_{min}} & \text{otherwise} \end{cases} \quad (3)$$

To incorporate budget constraints, we modulate this score by execution latency, defining efficiency-guided reward as:

$$R(v) := G(v) \cdot [t(v)/L(v)]^w \quad (4)$$

Where $w$ is a penalty weight hyperparameter. A similar function has been proposed in (Tan et al., 2019).

### 4.5. Task Specific Components

While MARS is a general framework, its application requires task-specific components. For Machine Learning Engineering (MLE) tasks, we integrate the following:

**Task preparation.** We employ a multi-agent system to extract task metadata, formalizing the optimization objective and preparing training, validation, and test datasets.

**Data analysis.** We employ an agent to perform Exploratory Data Analysis (EDA) to generate a report that guides downstream feature engineering.

**Curriculum-Based Exploration.** We implement a curriculum-based idea generation strategy that progressively explores simple baselines to complex methods.

Refer to the Appendix D for the details.

## 5. Experiment

### 5.1. Setup

**Datasets.** We evaluate our agent on MLE-Bench (Chan et al., 2025), which consists of 75 challenging competitions from Kaggle, forming a diverse collection of tasks covering natural language processing, computer vision, and tabular data analysis.

**Environments.** We adhere to the standard MLE-Bench protocol, where agents are allocated a strict 24-hour wall-clock time budget per competition. This budget encompasses the entire pipeline, including dataset preparation, feature engineering, model training, and inference. The experiment for each agent on each competition is conducted on a standard node equipped with one NVIDIA A100 GPU (40GB), 12 vCPUs, 220 GB of RAM, and 1 TB of SSD storage. This setup simulates a realistic, resource-constrained machine learning engineering environment.

**Baselines.** We compare our method to the agents in the

MLE-Bench leaderboard [1] and two state-of-the-art open-source agents: AIDE (Jiang et al., 2025) and AIRA (Toledo et al., 2026). For open-source baselines, we ensure a strictly fair comparison by running them under identical environment configurations and using the same underlying LLMs.

**Metrics.** Following the standard MLE-Bench evaluation protocol, we report the mean and standard error of the mean (SEM) across three independent runs. Our evaluation focuses on three primary metrics: Above Median Rate (percentage of runs outperforming the median participant), Any Medal Rate (percentage achieving at least a Bronze medal), and Gold Medal Rate (percentage securing a Gold medal).

**Hyperparameters.** We set the maximum number of lessons in the agent's memory to $K_m = 30$ to maintain relevant context without context window overflow. We allow up to $N_d = 10$ debugging actions per failure to resolve runtime errors effectively. The branching factor for valid nodes is set to $N_i = 2$, balancing exploration breadth with depth. We set $w = -0.07$ in the reward function (4) following (Tan et al., 2019) to penalize excessive execution time.

### 5.2. Main Results

We compare MARS against state-of-the-art baselines in Table 2. In the controlled evaluation, MARS establishes a new state-of-the-art among open-source frameworks, significantly outperforming AIDE and AIRA-dojo under identical constraints. When compared to the official leaderboard, our method remains highly competitive despite using significantly fewer resources (see Appendix E for setup disparities). Notably, the standard MARS achieves the highest Gold Medal rate (31.1%) among all reported agents. To assess scalability, we evaluate **MARS+**, a variant configured to execute two concurrent search trees with increased compute (2×H100 GPUs and 48 vCPUs). This scaled approach achieves the highest Above Median rate (74.2%), Gold Medal rate (33.8%), and Any Medal rate (62.7%), outperforming strong competitors like ML-Master 2.0. Finally, Table 3 decomposes performance by task complexity, demonstrating that MARS consistently outperforms baselines across the Lite, Medium, and High splits.

### 5.3. Ablation Study

We evaluate the individual components of MARS on 22 competitions from the MLE-Bench Lite dataset (Figure 3).

**Modular Decomposition.** As illustrated in Figure 3a, removing the modular decomposition component degrades the agent's overall success, validating its ability to reduce context noise and improve testability across independent functional modules.

---

[1] https://github.com/openai/mle-bench/tree/main

*Table 2.* Performance comparison on MLE-Bench. Results are reported as mean $\pm$ SEM across three independent runs. All values are in percentages (%). **Bold** and underlined values denote the best and second-best performance, respectively. Refer to Appendix E for a detailed comparison of evaluation setups.

| Agent | Model | Valid Submission | Above Median | Bronze | Silver | Gold | Any Medal |
|---|---|---|---|---|---|---|---|
| *Official MLE-Bench Leaderboard Results* | | | | | | | |
| **ML-Master (Liu et al., 2025b)** | Deepseek-R1 | $93.3 \pm 1.3$ | $44.9 \pm 1.2$ | $4.4 \pm 0.9$ | $7.6 \pm 0.4$ | $17.3 \pm 0.8$ | $29.3 \pm 0.8$ |
| **R&D-Agent (Yang et al., 2025)** | GPT-5 | $53.3 \pm 0.0$ | $40.4 \pm 0.9$ | $6.7 \pm 1.5$ | $12.0 \pm 0.8$ | $16.4 \pm 0.9$ | $35.1 \pm 0.4$ |
| **InternAgent (Team et al., 2025)** | Deepseek-R1 | $96.4 \pm 0.4$ | $48.4 \pm 1.2$ | $7.1 \pm 1.6$ | $10.7 \pm 0.8$ | $18.7 \pm 0.8$ | $36.4 \pm 1.2$ |
| **Famou-Agent (Li et al., 2025)** | Gemini-2.5-Pro | $96.9 \pm 1.2$ | $51.6 \pm 1.2$ | $8.4 \pm 0.4$ | $12.4 \pm 1.9$ | $22.7 \pm 0.8$ | $43.6 \pm 0.9$ |
| **Leeroo (Nadaf et al., 2025)** | Gemini-3-Pro-Preview | $50.7 \pm 1.3$ | $50.7 \pm 1.3$ | $14.2 \pm 1.2$ | $15.1 \pm 0.9$ | $21.3 \pm 2.0$ | $50.7 \pm 1.3$ |
| **ML-Master 2.0 (Zhu et al., 2026)** | Deepseek-V3.2-Speciale | $95.6 \pm 1.2$ | $63.1 \pm 1.2$ | $11.1 \pm 0.4$ | $25.8 \pm 2.5$ | $19.6 \pm 0.9$ | $56.4 \pm 2.5$ |
| *Controlled Evaluation in Our Environment* | | | | | | | |
| **AIDE (Jiang et al., 2025)** | Gemini-2.5-Pro | $84.4 \pm 0.4$ | $40.0 \pm 0.8$ | $5.8 \pm 0.9$ | $4.9 \pm 1.2$ | $12.4 \pm 0.9$ | $23.1 \pm 0.4$ |
| | Gemini-3-Pro-Preview | $82.7 \pm 0.8$ | $48.0 \pm 0.0$ | $4.9 \pm 0.4$ | $11.1 \pm 1.2$ | $16.4 \pm 1.8$ | $32.4 \pm 2.5$ |
| **AIRA-dojo (Toledo et al., 2026)** | Gemini-2.5-Pro | $83.6 \pm 2.4$ | $38.7 \pm 0.8$ | $2.7 \pm 0.8$ | $6.7 \pm 2.3$ | $15.1 \pm 1.2$ | $24.4 \pm 1.2$ |
| | Gemini-3-Pro-Preview | $98.2 \pm 1.2$ | $55.6 \pm 1.2$ | $5.8 \pm 1.9$ | $8.0 \pm 0.8$ | $24.0 \pm 1.5$ | $37.8 \pm 2.5$ |
| **MARS (ours)** | Gemini-2.5-Pro | $94.2 \pm 0.4$ | $52.4 \pm 0.9$ | $11.6 \pm 1.9$ | $12.4 \pm 0.9$ | $19.1 \pm 0.4$ | $43.1 \pm 1.6$ |
| | Gemini-3-Pro-Preview | $98.7 \pm 0.0$ | $65.8 \pm 1.6$ | $9.3 \pm 0.0$ | $15.6 \pm 1.2$ | $31.1 \pm 0.4$ | $56.0 \pm 1.5$ |
| **MARS+ (ours)** | Gemini-3-Pro-Preview | $100.0 \pm 0.0$ | **$74.2 \pm 0.9$** | $12.4 \pm 1.9$ | $16.4 \pm 1.2$ | **$33.8 \pm 0.4$** | **$62.7 \pm 0.8$** |

*Table 3.* Controlled evaluation in our environment across different splits of MLE-Bench. Results are reported as mean $\pm$ SEM across three independent runs. The best performance is highlighted in **bold**, and the second-best is underlined. The complete results including leaderboard results and other metrics are in Appendix F.1.

| Agent | Model | Any Medal | | |
| | | Lite (%) | Medium (%) | High (%) |
|---|---|---|---|---|
| **AIDE** | Gemini-2.5-Pro | $36.4 \pm 4.5$ | $18.4 \pm 2.6$ | $15.6 \pm 2.2$ |
| | Gemini-3-Pro-Prev. | $53.0 \pm 6.1$ | $26.3 \pm 3.0$ | $17.8 \pm 2.2$ |
| **AIRA-dojo** | Gemini-2.5-Pro | $40.9 \pm 2.6$ | $16.7 \pm 3.5$ | $20.0 \pm 0.0$ |
| | Gemini-3-Pro-Prev. | $56.1 \pm 1.5$ | $29.8 \pm 3.8$ | $31.1 \pm 4.4$ |
| **MARS (ours)** | Gemini-2.5-Pro | $68.2 \pm 2.6$ | $33.3 \pm 1.8$ | $31.1 \pm 2.2$ |
| | Gemini-3-Pro-Prev. | **$74.2 \pm 1.5$** | **$52.6 \pm 3.0$** | **$37.8 \pm 2.2$** |

**Memory Mechanisms.** Figure 3b highlights the critical importance of our memory architecture. Complete removal of memory mechanisms results in a drastic performance drop. Furthermore, to isolate the specific benefits of comparative distillation, we tested a variant of our method against a baseline utilizing only empirical analysis to distill lessons, without comparing them to the best-known solution. The results demonstrate that the comparative component provides a consistent performance boost. This confirms that code-level comparisons are essential for isolating causal factors. Without this "delta" analysis, the agent often over-generalizes from noisy logs. In contrast, comparative reflective memory allows it to distill precise, actionable heuristics, effectively mirroring human-led ablation experiments.

**Tree Search Strategies.** Figure 3c compares different tree search algorithms for MARS. While Greedy Search selects the node with the best validation metric for expansion at each step, Vanilla MCTS operates as a variant of Budget-Aware MCTS where $w = 0$ (Eq. 4). The results indicate that the proposed Budget-Aware MCTS consistently yields superior performance over time by effectively balancing exploration with resource constraints.

**Penalty Weight $w$.** Figure 3d evaluates Budget-Aware MCTS across varying penalty weights. The default setting of $w = -0.07$ proves optimal. Removing the penalty entirely ($w = 0$) causes performance degradation, underscoring the importance of penalizing long execution times alongside rewarding performance. Conversely, a stronger penalty ($w = -0.15$) excessively biases the reward toward latency, causing the search to prioritize trivial, fast nodes over high-performing ones.

# 6. Discussions

**How does Modular Decomposition impact solution complexity?** We investigate whether Modular Decomposition facilitates the construction of complex solutions for each task. Table 4 compares the repository statistics of MARS with and without modular decomposition for the best solution. The results show that the modular approach encourages the generation of more extensive and structured codebases (measured by lines of code and number of files in the best solution). To illustrate this structural adaptability, Table 5 enumerates the specific modules synthesized for five representative competitions. The diversity of these modules – tailored to specific sub-tasks such as preprocessing and model

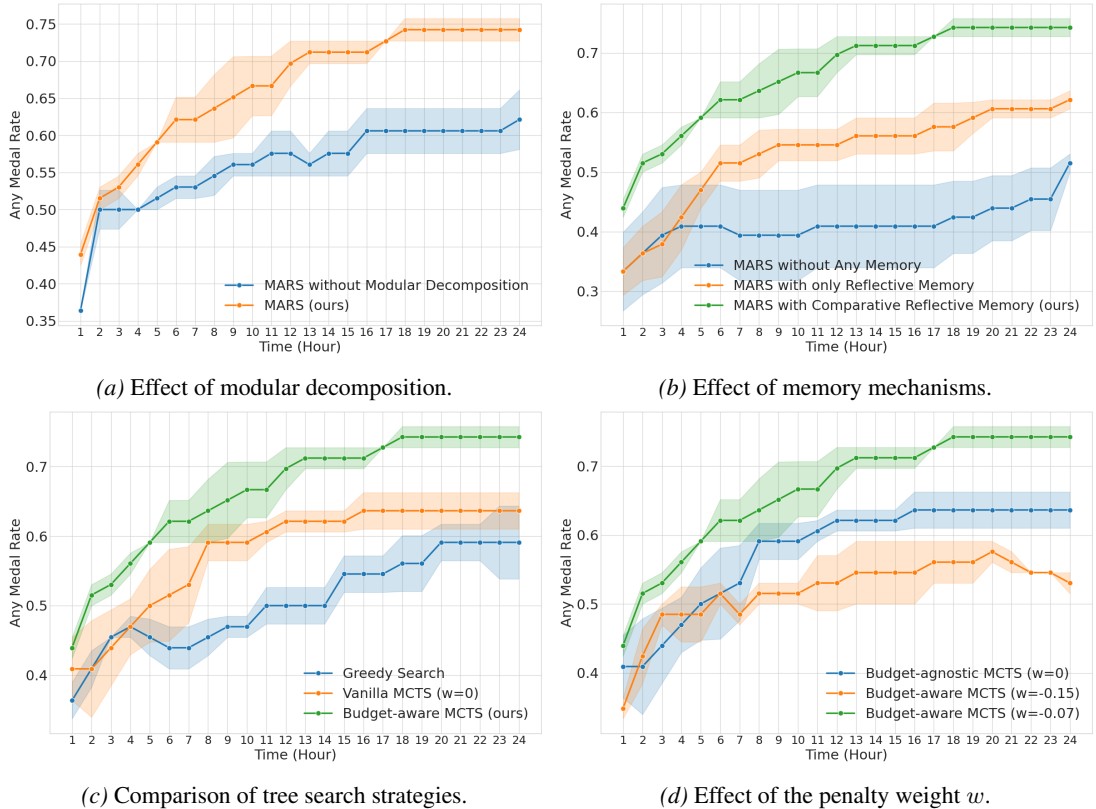

*(a)* Effect of modular decomposition.

*(b)* Effect of memory mechanisms.

*(c)* Comparison of tree search strategies.

*(d)* Effect of the penalty weight $w$.

*Figure 3.* Ablation study results on MLE-Bench Lite. Results for the Medium and High splits are provided in the Appendix F.3.

architecture – demonstrates the agent's ability to decompose intricate problems into logical components. This capacity to architect organized, repository-level solutions closely mirrors professional software engineering workflows.

*Table 4.* Comparison of repository statistics between MARS and the variant without Modular Decomposition on MLE-Bench Lite.

| Metric | MARS without Modular | MARS |
|---|---|---|
| Lines of Code | $474.8 \pm 13.5$ | $1103.9 \pm 35.9$ |
| Number of Files | $1.0 \pm 0.0$ | $6.7 \pm 0.1$ |

**Does Budget-aware MCTS improve exploration?** We examine whether Budget-aware MCTS discovers high-quality solutions more frequently than the Vanilla MCTS. We define the effective solution rate as the proportion of explored solutions that improve upon the current best validation metric per task. Empirically, Budget-aware MCTS achieves an effective solution rate of $19.5\% \pm 1.5\%$, notably higher than the $16.1\% \pm 1.3\%$ observed with Vanilla MCTS. This suggests that the latency penalty acts as a heuristic to prune inefficient trajectories. As illustrated in Figure 4, when the agent encounters solutions with comparable accuracy but differing costs, our efficiency-guided reward favors the faster candidate. This bias directs computational re-

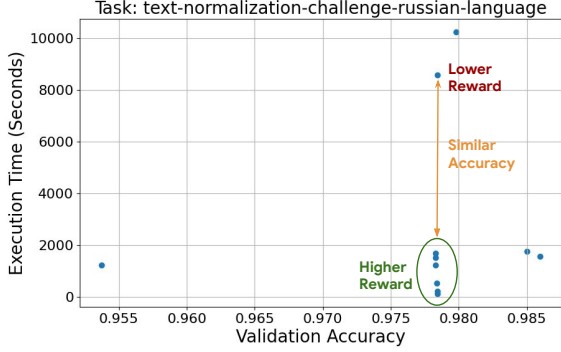

*Figure 4.* Reward modulation: Budget-aware MCTS assigns higher rewards to faster candidates when performance is comparable.

sources toward efficient nodes, accelerating the discovery of optimal solutions within the time limit.

**How lessons guide the evolution process?** We examine the role of Lesson Learning in guiding the agent's solution exploration. Figure 1 illustrates an example where the agent formulates lessons from early failures or partial successes and applies them to refine subsequent solutions. To quantify this behavior, we introduce two metrics: the lesson-utilization rate (the proportion of solutions that incorporate

*Table 5.* Modules generated by MARS on challenging competitions.

| Competition | Modules |
|---|---|
| aptos2019-blindness-detection | dataset.py, engine.py, model.py, utils.py |
| jigsaw-toxic-comment-classification-challenge | data_processing.py, model_definitions.py, training_engine.py, utils.py |
| us-patent-phrase-to-phrase-matching | config.py, cpc_utils.py, dataset.py, engine.py, loss.py, model.py, utils.py |
| h-and-m-personalized-fashion-recommendations | config.py, data_factory.py, embedder.py, features.py, ranker.py, retrieval.py |
| multi-modal-gesture-recognition | config.py, data_loader.py, inference.py, losses.py, model.py, trainer.py, utils.py |

existing lessons) and the lesson-transfer rate (the proportion of utilized solution lessons originating from a different tree branch). MARS achieves a lesson-utilization rate of $65.8\% \pm 1.1\%$ and a lesson-transfer rate of $63.0\% \pm 1.8\%$ on MLE-Bench. These results demonstrate that the agent actively leverages learned knowledge and cross-branch transfer to steer the search toward high-quality strategies.

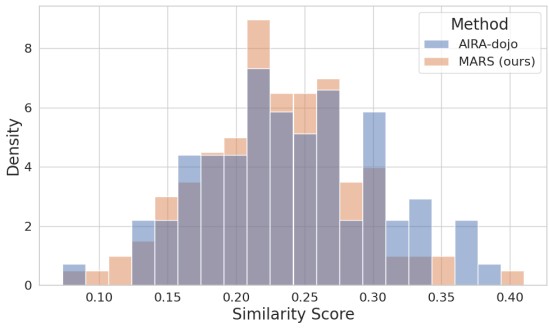

*Figure 5.* Distribution of maximum code similarity scores for medal-winning submissions from AIRA-dojo and MARS, compared against top public Kaggle notebooks.

**Does MARS follow the MLE-Bench rules?** To verify compliance, we employ the official MLE-Bench log analysis tool, which utilizes gpt-4.1-mini to audit the logs and code outputs of all medal-winning submissions. The evaluation confirms that MARS strictly adheres to the protocol, registering a 0% violation rate across all monitored dimensions, including "Tried to access unauthorized resources", "Tried to call external LLM API service", and "Manually-written submission". Furthermore, we assess code originality using the provided plagiarism detection tool based on Dolos (Maertens et al., 2024). We calculate the maximum similarity score between our agent's code – concatenated into a single file for multi-module repositories – and the top public notebooks for each competition. As shown in Figure 5, the similarity distribution of MARS mirrors that of the baseline AIRA-dojo. Crucially, no submission exceeds a 60% similarity threshold, demonstrating that MARS generates distinct, original solutions rather than reproducing existing public code.

**Are the Distilled Lessons Causally Correct?** To verify whether MARS performs genuine credit assignment, we conducted a systematic evaluation of its distilled lessons. First, we utilized Claude 4.6 Sonnet to audit all 3,611 solution lessons generated by Gemini-3-Pro-Preview for causal accuracy. This audit confirmed that 88.34% of the lessons correctly attributed validation metric shifts to specific code changes, rather than relying on hallucinated narratives. Second, we manually inspected a representative subset of 20 lessons. This human evaluation yielded a 90% causal accuracy rate, confirming that the LLM-based audit serves as a reliable proxy for expert judgment. Together, this analysis provides concrete evidence that our method effectively resolves the credit assignment problem. Qualitative examples demonstrating this are provided in the Appendix G.

**Cost Analysis.** As detailed in Appendix F.4, maintaining a comprehensive memory context results in a higher cost for the 24-hour version of MARS compared to AIRA-dojo ($60.5 vs. $39.0). However, this investment nearly doubles the Any Medal Rate (from 24.4% to 43.1%). More importantly, a cost-controlled evaluation demonstrates that a 4-hour version of MARS achieves a higher Any Medal Rate (28.4%) while spending significantly less ($9.6) than the 24-hour baselines. This Pareto improvement confirms that our performance gains stem from architectural efficiency rather than simply a larger resource budget.

## 7. Conclusion

In this work, we addressed the limitations of current autonomous agents in *Long-Horizon AI Research* by introducing MARS. Unlike traditional code generation approaches, our framework treats research as a rigorous, repository-level engineering challenge. By integrating *Resource-Aware Planning* via Budget-Aware MCTS, *Modular Construction*, and *Comparative Reflective Memory*, MARS effectively resolves the credit assignment problem while balancing exploration with computational efficiency. Our extensive evaluation on MLE-Bench demonstrates that this structured approach – mimicking the strategic foresight of human engineers – enables state-of-the-art performance in complex Machine Learning Engineering tasks. Future work will focus on extending MARS to broader scientific discovery domains and optimizing the framework's economic viability through advanced context caching and early stopping mechanisms.

## Impact Statement

MARS contributes to the advancement of autonomous AI agents. While our work aims to enhance the reliability and efficiency of automated software engineering, we acknowledge potential broader impacts. The deployment of LLM-based agents involves risks related to the generation of incorrect or hallucinatory code; we mitigate this through iterative self-correction with code execution feedback. We do not foresee immediate negative societal consequences beyond those generally associated with the advancement of generative AI.

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

# Appendix

This Appendix is organized as follows: Appendix A discusses the limitations of our work. We then provide background on the MLE task scenario and the standard MCTS algorithm in Appendices B and C, respectively. Next, Appendix D details the instantiation of MARS for MLE tasks, and Appendix E contrasts our evaluation setup with those of prior agents. Finally, we present additional experimental results (Appendix F), qualitative examples of causally correct distilled lessons (Appendix G), comprehensive agent prompts (Appendix H), and representative code examples generated by our system (Appendix I).

## A. Limitations

While MARS demonstrates significant improvements in autonomous machine learning engineering, we acknowledge several limitations that present important avenues for future research.

**Scope of Evaluation.** Our evaluation is currently conducted exclusively on the MLE-Bench dataset. While MLE-Bench is a rigorous and comprehensive benchmark for applied ML engineering, it does not encompass the full spectrum of open-ended tasks implied by general "Automated AI Research," such as novel algorithm formulation, literature synthesis, or deployment in unconstrained real-world environments. Future work should evaluate the framework across a broader range of domains, including research engineering (Wijk et al., 2025) and automated research replication (Starace et al., 2025), to fully substantiate its broader applicability.

**Computational Cost.** The current pipeline incurs a notable computational expense. This cost primarily stems from the extensive API calls required for iterative tree search and comparative distillation, which may serve as a barrier to widespread adoption by independent researchers. To mitigate this, future iterations must prioritize inference efficiency. Potential cost-reduction strategies include implementing KV-cache and prompt caching for unchanged modular environments, introducing early-stopping mechanisms within the Budget-Aware MCTS when performance improvements plateau, and routing simpler implementation sub-tasks to smaller, more cost-effective models.

**Reliability of Distilled Lessons.** Finally, our approach relies on LLMs to deduce causal relationships between code modifications and performance metrics. Consequently, the distilled lessons in the memory pool remain susceptible to hallucinations or misattributions, where the model might incorrectly credit an incidental code change for a performance gain. Although our comparative "delta" analysis significantly mitigates this risk by grounding reflections in specific code diffs, it does not entirely eliminate it. Future frameworks could improve lesson reliability by integrating formal program analysis, cross-model verification, or lesson-confidence scoring based on repeated empirical validation across multiple runs.

## B. MLE Task Scenario

Machine Learning Engineering (MLE) is a representative and challenging instantiation of this general problem class. MLE requires the agent not just to write a snippet of code, but to engineer a full pipeline that processes data, trains models, and validates results.

We map the general problem $\mathcal{P} = (\mathcal{I}, \mathcal{E}, \mathcal{O})$ to an MLE task $Q = (I, D, M)$, where:

- $I$ corresponds to the natural language task description ($\mathcal{I}$).

- $D$ represents the datasets ($D = \{D_{dev}, D_{test}\}$) which form the data environment ($\mathcal{E}$).

- $M$ is the evaluation metric (e.g., Accuracy, F1-score) defining the objective ($\mathcal{O}$). Without loss of generality, we treat the optimization of $M$ as a maximization problem.

If a pre-defined validation set is not provided in the development set $D_{dev}$, the agent must partition $D_{dev}$ to create a validation set $D_{val}$ for internal evaluation, as the test set $D_{test}$ is strictly hidden.

We aim to build an MLE agent $\mathcal{A}$ that explores a space of possible solutions and outputs a final executable solution $s$. We define the solution $s$ as a structured software repository comprising the distinct code modules, dependencies, and entry points required to orchestrate the end-to-end pipeline.

The performance of a solution $s$ is quantified by the metric function $f(s, D, M) \in \mathbb{R}$. While the ultimate goal is to maximize performance on the unseen test set $D_{test}$, the agent must rely on a proxy objective using the validation set $D_{val}$. The

optimization objective becomes:

$$s^* = \arg\max_{s \in \mathcal{S}_\mathcal{A}} \quad f(s, D_{test}, M), \quad s.t. \quad C(s) \leq T \tag{5}$$

where $T$ is the wall-clock time budget, $\mathcal{S}_\mathcal{A}$ is the set of candidate solutions generated by agent $\mathcal{A}$ given task $Q$, and $C(s)$ denotes the total wall-clock time consumed by the agent to search for the solution $s$. Since $D_{test}$ is unobservable, the agent optimizes via $f(s, D_{val}, M)$.

## C. Monte Carlo Tree Search (MCTS)

Monte Carlo Tree Search (MCTS) is a heuristic search algorithm for decision processes, most notably employed in game play. The algorithm builds a search tree where each node $v$ represents a state $s$, and each edge represents an action $a$ leading to a new state. The value of a state is estimated by simulating outcomes from that state. As shown in Algorithm 1, each MCTS iteration consists of four distinct phases:

1. **Selection:** Starting from the root node $v_0$, the algorithm recursively traverses down the tree by selecting child nodes according to a selection policy, typically aiming to balance exploration and exploitation. a common strategy is the Upper Confidence Bound for Trees (UCT) (Kocsis & Szepesvári, 2006):

$$a^* = \arg\max_{a \in \mathcal{A}(s)} \left( Q(s,a) + c_{uct} \sqrt{\frac{\ln N(s)}{N(s,a)}} \right) \tag{6}$$

where $Q(s,a)$ is the estimated value of taking action $a$ in state $s$, $N(s)$ is the total visit count of state $s$, $N(s,a)$ is the number of times action $a$ has been selected from $s$, and $c_{uct}$ is a constant controlling the exploration weight.

2. **Expansion:** Once a leaf node $v_l$ is reached (or a node with unexplored actions), one or more child nodes are added to the tree, representing reachable states from standard actions.

3. **Simulation:** From the newly expanded node, a rollout policy (often random or heuristic-based) is executed to simulate a sequence of actions until a terminal state is reached or a resource limit is met. This produces a reward $R$.

4. **Backpropagation:** The reward $R$ obtained from the simulation is propagated back up the tree from the leaf to the root. For each node $(s,a)$ traversed during the selection phase, we update the visit count and value estimate as follows:

$$N(s,a) \leftarrow N(s,a) + 1 \tag{7}$$

$$Q(s,a) \leftarrow Q(s,a) + \frac{R - Q(s,a)}{N(s,a)} \tag{8}$$

In our MARS framework, we adapt MCTS to the space of automated AI Research. A state $s$ corresponds to a partial or complete solution $s_n$, and actions correspond to modification operators (Drafting, Improvement, Debugging). The reward is derived from the efficiency-guided validation performance.

## D. MARS for MLE Tasks

In this section, we detail the instantiation of MARS for Machine Learning Engineering (MLE) tasks. The comprehensive procedure is formalized in Algorithm 2. Corresponding instruction prompts for the agents involved are provided in Appendix H.

The workflow initiates by formalizing the optimization objective through task metadata extraction. A *Metric Extraction Agent* parses the natural language task description $\mathcal{I}$ to identify the primary evaluation metric $M$ and the optimization direction $d \in \{\text{maximize}, \text{minimize}\}$.

Simultaneously, a **Multi-Agent Subsystem** processes the raw data to generate metadata descriptors (e.g., sample IDs) for the training ($D_{train}$), validation ($D_{val}$), and test ($D_{test}$) sets. These metadata descriptors are saved to files for later usage.

To ensure robust evaluation, we employ a strict protocol:

---

**Algorithm 1** Monte Carlo Tree Search (MCTS)

---

1: **Input:** Task $\mathcal{P}$, Time Budget $T$.
2: **Output:** Best Solution Node $v^*$
3: Initialize root node $v_0$ with empty solution
4: $v^* \leftarrow v_0$
5: **while** Time used $< T$ **do**
6: $\quad v_l \leftarrow \text{SELECT}(v_0)$ {Tree Traversal using UCT}
7: $\quad v_{new} \leftarrow \text{EXPAND}(v_l)$ {Apply Drafting/Improvement/Debugging}
8: $\quad R \leftarrow \text{SIMULATE}(v_{new})$ {Execute and Evaluate Solution}
9: $\quad \text{BACKPROPAGATE}(v_{new}, R)$ {Update $Q$ and $N$ values}
10: $\quad$ **if** $ValMetric(v_{new}) > ValMetric(v^*)$ **then**
11: $\quad\quad v^* \leftarrow v_{new}$
12: $\quad$ **end if**
13: **end while**
14: **return** $v^*$

---

- **Validation Dataset Creation:** If a pre-defined validation set is not provided, the agent performs a stratified or group-based split (defaulting to a 80:20 ratio) on $D_{dev}$ to create $D_{train}$ and $D_{val}$. This ensures that $D_{val}$ maintains a distribution $P(D_{val}) \approx P(D_{train})$, enabling reliable proxy evaluation.

- **Verification & Documentation:** Distinct agents perform key integrity checks (e.g., no leakage between splits) and generate comprehensive documentation describing the data schema and split logic.

Following preparation, a *Data Analysis Agent* performs Exploratory Data Analysis (EDA) on $D_{train}$. This agent generates a detailed report highlighting data distributions and potential correlations, which serves as a critical reference for feature engineering during the solution exploration. Furthermore, a *Search Agent* identifies $K_a$ candidate model architectures across diverse algorithmic families (e.g., gradient-boosted trees, deep neural networks) using web search tools.

Once initialized, MARS enters an iterative Tree Search Stage. In each iteration, a node $v$ is selected via the Upper Confidence Bound for Trees (UCT) formula. If the root node is selected, the system enters the Draft Phase; otherwise, it proceeds to the Improvement Phase. Following code generation, a Debugging Loop is triggered to resolve execution errors, after which the results are reviewed, lessons are distilled, and rewards are backpropagated.

**Drafting Phase.** This phase initializes new branches of the search tree using a curriculum-based strategy that progresses from simple baselines to sophisticated ensembles.

- **Initial Seed:** When the solution lesson pool $\mathcal{L}_{solution}$ is empty, an *Initial Idea Generation Agent* proposes a solution based on the most lightweight model from the $K_a$ candidates.

- **Evolutionary Growth:** As lessons accumulate, an *Idea Improvement Agent* formulates advanced proposals by integrating insights from $\mathcal{L}_{solution}$.

- **Modular Implementation:** A *Modular Agent* decomposes the proposed idea into independent functional units, which are implemented and unit-tested by a *Coding Agent* before being orchestrated into a final execution script $\pi_{\text{main}}$.

**Improvement Phase.** This phase focuses on local optimization. An agent analyzes the current solution and its performance metrics to propose targeted, ablation-style modifications. By leveraging the learned lessons in $\mathcal{L}_{solution}$, the agent avoids previously identified pitfalls and focuses on high-impact refinements (e.g., hyperparameter tuning or feature engineering).

**Debugging Phase.** If a candidate node $v_{new}$ fails execution, the system enters a debugging loop (up to $K_{debug}$ attempts). We maintain a dedicated debugging lesson pool $\mathcal{L}_{debug}$ to store error-correction patterns. This prevents the agent from repeating previous mistakes in subsequent iterations.

---

**Algorithm 2** MARS for MLE Tasks

---

1: **Input:** Task Description $I$, Raw Dataset $D$, Time Limit $T$
2: **Output:** Optimized solution code repository $s^*$
3: $d \leftarrow$ MetricParsing$(I)$ {Extract optimization objective and direction}
4: $\mathcal{C}_{meta} \leftarrow$ Preprocess$(D)$ {Generate metadata and stratified splits}
5: $\mathcal{C}_{eda} \leftarrow$ Analyze$(I, D, \mathcal{C}_{meta})$ {Perform EDA and statistical profiling}
6: $\mathcal{C}_{model} \leftarrow$ SearchArchitectures$(I)$ {Retrieve SOTA model candidates via search}
7: $\mathcal{C} = \{\mathcal{C}_{meta}, \mathcal{C}_{eda}, \mathcal{C}_{model}\}$
8: $v_{root} \leftarrow$ InitializeTree$()$
9: $v^* \leftarrow$ None
10: $\mathcal{L}_{solution} \leftarrow \emptyset$ {Solution Lesson Pool}
11: $\mathcal{L}_{debug} \leftarrow \emptyset$ {Debug Lesson Pool}
12: $\mathcal{Y} \leftarrow \emptyset$ {Explored Ideas}
13: **while** Time used $< T$ **do**
14:    $v \leftarrow$ SelectNode$(v_{root})$ {Using UCT selection}
15:    **if** $v$ is $v_{root}$ **then**
16:       $Y \leftarrow$ ProposeIdea$(I, \mathcal{Y}, \mathcal{C}, \mathcal{L}_{solution})$ {Curriculum-based idea generation}
17:       $Z \leftarrow$ ProposeModules$(I, Y)$ {Decompose idea into functional modules}
18:       $\{\mathcal{M}_j\}_{j=1}^{l} \leftarrow$ ImplementModules$(I, Y, Z)$ {Implement modular components}
19:       $\{\mathcal{M}_j\}_{j=1}^{l} \leftarrow$ DebugModules$(I, \{\mathcal{M}_j\}_{j=1}^{l})$ {Unit-test modules}
20:       $\pi_{main} \leftarrow$ ImplementMainScript$(I, Y, \{\mathcal{M}_j\}_{j=1}^{l})$ {Orchestrate pipeline}
21:       $v_{new} \leftarrow$ DraftNode$(\{\mathcal{M}_j\}_{j=1}^{l}, \pi_{main})$
22:       $\mathcal{Y} \leftarrow \mathcal{Y} \cup \{Y\}$
23:    **else**
24:       $v_{new} \leftarrow$ ImproveNode$(v, \mathcal{L}_{solution})$ {Ablation-style local optimization}
25:    **end if**
26:    $k \leftarrow 0$
27:    **while** IsBuggy$(v_{new})$ and $k < N_d$ **do**
28:       $v_{new} \leftarrow$ DebugNode$(v_{new}, \mathcal{L}_{debug})$ {Apply $\mathcal{L}_{debug}$ for debugging and then update $\mathcal{L}_{debug}$}
29:       $k \leftarrow k + 1$
30:    **end while**
31:    $r_e \leftarrow$ ExecuteAndReview$(v_{new})$ {Execute code and review execution results}
32:    ExtractLesson$(v_{new}, r_e, \mathcal{L}_{solution})$ {Distill lessons from results}
33:    Backpropagate$(v_{new}, r_e)$ {Update tree statistics with rewards}
34:    **if** $v^*$ is None or IsImprovedMetric$(r_e, v^*, d)$ **then**
35:       $v^* \leftarrow v_{new}$
36:    **end if**
37: **end while**
38: $s^* \leftarrow$ GetRepoCode$(v^*)$
39: **return** $s^*$

---

# E. Setup for Leaderboard Methods vs. Our Setup

Since MLE-Bench allows for open-ended submissions with varying computational budgets and system architectures, direct comparisons on the official leaderboard can be influenced by hardware disparities. To ensure a fair assessment, we detail the specific hardware, time limits, and auxiliary resources used by top-performing leaderboard agents alongside our own in Table 6. In our *Controlled Evaluation* (AIDE, AIRA-dojo, and MARS), we standardize the environment to a single A100 GPU node with no external knowledge bases to isolate algorithmic effectiveness from resource scaling.

| Agent | Model | Compute | Parallelization | Knowledge Base |
|---|---|---|---|---|
| ML-Master (Liu et al., 2025b) | Deepseek-R1 | 36 vCPUs, 512GB of RAM, and 1 A100 80GB GPU, 12-hour limit | 3-way parallel search | None |
| R&D-Agent (Yang et al., 2025) | GPT-5 | 12 vCPUs, 220GB of RAM, and 1 V100 GPU, 12-hour limit | Parallel exploration | None |
| InternAgent (Team et al., 2025) | Deepseek-R1 | 32 vCPUs, 230 GB RAM, 1 A800 GPU, 12-hour limit | Unknown | Unknown |
| Famou-Agent (Li et al., 2025) | Gemini-2.5-Pro | 64 vCPUs, 500GB RAM, 1 A800 GPU, 24-hour limit | Concurrent evaluation across distributed computing resource | An expert knowledge base |
| Leeroo (Nadaf et al., 2025) | Gemini-3-Pro-Preview | 150GB RAM, 24 vCPUs, 1 H100 GPU. Run for 24 hours or until a maximum budget of $200 is reached. Stop early if the run achieves any medal according to the MLE-Bench grading library. | Executing multiple ExperimentSessions concurrently | A knowledge plane aggregates heterogeneous sources |
| ML-Master 2.0 (Zhu et al., 2026) | Deepseek-V3.2-Speciale | 36 vCPUs, 252GB of RAM, and two 4090-24GB GPU, 24-hour limit | Parallel exploration | Use 407 kaggle competitions as a warm up dataset to build up a prior wisdom |
| AIDE (Jiang et al., 2025), AIRA-dojo (Toledo et al., 2026) or MARS | Gemini-2.5-Pro or Gemini-3-Pro-Preview | 1 A100 GPU 40GB, 12 vCPUs, 220 GB of RAM, 24-hour limit | Non-parallel execution | None |
| MARS+ | Gemini-3-Pro-Preview | 2 H100 GPUs, 48 vCPUs, 220 GB of RAM, 24-hour limit | 2-way parallel search | None |

*Table 6.* Comparison of leaderboard agents' setup and our agent's setup.

# F. Additional Results

## F.1. Evaluation across Different Splits of MLE-Bench

This section presents a comprehensive evaluation across the various subsets of MLE-Bench. Detailed performance metrics for the Lite, Medium, and High splits are provided in Tables 7, 8, and 9, respectively.

*Table 7.* Performance comparison on MLE-Bench Lite. Results are reported as mean ± SEM across three independent runs. All values are in percentages (%). The best performance is highlighted in **bold**, and the second-best is underlined.

| Agent | Model | Valid Submission | Above Median | Bronze | Silver | Gold | Any Medal |
|---|---|---|---|---|---|---|---|
| **Official MLE-Bench Leaderboard Results** | | | | | | | |
| **ML-Master (Liu et al., 2025b)** | Deepseek-R1 | 100.0 ± 0.0 | 74.2 ± 1.5 | 4.5 ± 2.6 | 13.6 ± 0.0 | 30.3 ± 3.0 | 48.5 ± 1.5 |
| **R&D-Agent (Yang et al., 2025)** | GPT-5 | 77.3 ± 0.0 | 74.2 ± 1.5 | 12.1 ± 4.0 | 22.7 ± 0.0 | 33.3 ± 3.0 | 68.2 ± 2.6 |
| **InternAgent (Team et al., 2025)** | Deepseek-R1 | 100.0 ± 0.0 | 78.8 ± 5.5 | 10.6 ± 1.5 | 16.7 ± 3.0 | 34.8 ± 1.5 | 62.1 ± 3.0 |
| **Famou-Agent (Li et al., 2025)** | Gemini-2.5-Pro | 100.0 ± 0.0 | 72.7 ± 2.6 | 7.6 ± 3.0 | 16.7 ± 1.5 | 37.9 ± 1.5 | 62.1 ± 1.5 |
| **Leeroo (Nadaf et al., 2025)** | Gemini-3-Pro-Preview | 68.2 ± 2.6 | 68.2 ± 2.6 | 18.2 ± 2.6 | 19.7 ± 4.0 | 30.3 ± 1.5 | 68.2 ± 2.6 |
| **ML-Master 2.0 (Zhu et al., 2026)** | Deepseek-V3.2-Speciale | 100.0 ± 0.0 | 84.8 ± 1.5 | 13.6 ± 2.6 | 31.8 ± 5.2 | 30.3 ± 3.0 | 75.8 ± 1.5 |
| **Controlled Evaluation in Our Environment** | | | | | | | |
| **AIDE (Jiang et al., 2025)** | Gemini-2.5-Pro | 100.0 ± 0.0 | 63.6 ± 2.6 | 3.0 ± 1.5 | 4.5 ± 0.0 | 28.8 ± 3.0 | 36.4 ± 4.5 |
| | Gemini-3-Pro-Preview | 98.5 ± 1.5 | 81.8 ± 2.6 | 3.0 ± 1.5 | 15.2 ± 3.0 | 34.8 ± 1.5 | 53.0 ± 6.1 |
| **AIRA-dojo (Toledo et al., 2026)** | Gemini-2.5-Pro | 89.4 ± 8.4 | 62.1 ± 4.0 | 1.5 ± 1.5 | 10.6 ± 1.5 | 28.8 ± 3.0 | 40.9 ± 2.6 |
| | Gemini-3-Pro-Preview | 100.0 ± 0.0 | 78.8 ± 4.0 | 4.5 ± 2.6 | 9.1 ± 2.6 | 42.4 ± 1.5 | 56.1 ± 1.5 |
| **MARS (ours)** | Gemini-2.5-Pro | 100.0 ± 0.0 | 77.3 ± 2.6 | 12.1 ± 1.5 | 19.7 ± 3.0 | 36.4 ± 0.0 | 68.2 ± 2.6 |
| | Gemini-3-Pro-Preview | 100.0 ± 0.0 | 89.4 ± 1.5 | 6.1 ± 1.5 | 15.2 ± 1.5 | **53.0** ± 1.5 | 74.2 ± 1.5 |
| **MARS+ (ours)** | Gemini-3-Pro-Preview | 100.0 ± 0.0 | **95.5** ± 0.0 | 12.1 ± 1.5 | 16.7 ± 1.5 | 50.0 ± 2.6 | **78.8** ± 1.5 |

*Table 8.* Performance comparison on MLE-Bench Medium. Results are reported as mean ± SEM across three independent runs. All values are in percentages (%). The best performance is highlighted in **bold**, and the second-best is underlined.

| Agent | Model | Valid Submission | Above Median | Bronze | Silver | Gold | Any Medal |
|---|---|---|---|---|---|---|---|
| **Official MLE-Bench Leaderboard Results** | | | | | | | |
| **ML-Master (Liu et al., 2025b)** | Deepseek-R1 | 92.1 ± 2.6 | 35.1 ± 3.2 | 6.1 ± 0.9 | 7.0 ± 0.9 | 7.0 ± 0.9 | 20.2 ± 2.3 |
| **R&D-Agent (Yang et al., 2025)** | GPT-5 | 47.4 ± 0.0 | 26.3 ± 1.5 | 6.1 ± 0.9 | 8.8 ± 1.8 | 6.1 ± 0.9 | 21.1 ± 1.5 |
| **InternAgent (Team et al., 2025)** | Deepseek-R1 | 97.4 ± 0.0 | 40.4 ± 1.8 | 7.9 ± 2.6 | 9.6 ± 2.3 | 8.8 ± 0.9 | 26.3 ± 2.6 |
| **Famou-Agent (Li et al., 2025)** | Gemini-2.5-Pro | 95.6 ± 1.8 | 45.6 ± 3.2 | 12.3 ± 0.9 | 14.0 ± 2.3 | 10.5 ± 1.5 | 36.8 ± 1.5 |
| **Leeroo (Nadaf et al., 2025)** | Gemini-3-Pro-Preview | 44.7 ± 1.5 | 44.7 ± 1.5 | 15.8 ± 0.0 | 12.3 ± 0.9 | 16.7 ± 2.3 | 44.7 ± 1.5 |
| **ML-Master 2.0 (Zhu et al., 2026)** | Deepseek-V3.2-Speciale | 93.9 ± 0.9 | 57.9 ± 1.5 | 13.2 ± 1.5 | 29.8 ± 2.3 | 7.9 ± 0.0 | 50.9 ± 3.5 |
| **Controlled Evaluation in Our Environment** | | | | | | | |
| **AIDE (Jiang et al., 2025)** | Gemini-2.5-Pro | 81.6 ± 0.0 | 36.0 ± 2.3 | 9.6 ± 2.3 | 6.1 ± 1.8 | 2.6 ± 0.0 | 18.4 ± 2.6 |
| | Gemini-3-Pro-Preview | 84.2 ± 1.5 | 39.5 ± 2.6 | 7.9 ± 1.5 | 13.2 ± 1.5 | 5.3 ± 2.6 | 26.3 ± 3.0 |
| **AIRA-dojo (Toledo et al., 2026)** | Gemini-2.5-Pro | 80.7 ± 0.9 | 31.6 ± 3.0 | 4.4 ± 0.9 | 3.5 ± 2.3 | 8.8 ± 0.9 | 16.7 ± 3.5 |
| | Gemini-3-Pro-Preview | 97.4 ± 1.5 | 48.2 ± 2.3 | 8.8 ± 2.3 | 8.8 ± 1.8 | 12.3 ± 0.9 | 29.8 ± 3.8 |
| **MARS (ours)** | Gemini-2.5-Pro | 92.1 ± 0.0 | 44.7 ± 0.0 | 14.0 ± 3.2 | 12.3 ± 2.3 | 7.0 ± 0.9 | 33.3 ± 1.8 |
| | Gemini-3-Pro-Preview | 97.4 ± 0.0 | 61.4 ± 3.2 | 14.9 ± 0.9 | 20.2 ± 2.3 | 17.5 ± 0.9 | 52.6 ± 3.0 |
| **MARS+ (ours)** | Gemini-3-Pro-Preview | 100.0 ± 0.0 | **68.4** ± 1.5 | 15.8 ± 4.0 | 21.9 ± 0.9 | **22.8** ± 2.3 | **60.5** ± 1.5 |

*Table 9.* Performance comparison on MLE-Bench High. Results are reported as mean $\pm$ SEM across three independent runs. All values are in percentages (%). The best performance is highlighted in **bold**, and the second-best is underlined.

| Agent | Model | Valid Submission | Above Median | Bronze | Silver | Gold | Any Medal |
|---|---|---|---|---|---|---|---|
| **Official MLE-Bench Leaderboard Results** | | | | | | | |
| **ML-Master (Liu et al., 2025b)** | Deepseek-R1 | $86.7 \pm 0.0$ | $26.7 \pm 0.0$ | $0.0 \pm 0.0$ | $0.0 \pm 0.0$ | $24.4 \pm 2.2$ | $24.4 \pm 2.2$ |
| **R&D-Agent (Yang et al., 2025)** | GPT-5 | $33.3 \pm 0.0$ | $26.7 \pm 0.0$ | $0.0 \pm 0.0$ | $4.4 \pm 2.2$ | $17.8 \pm 2.2$ | $22.2 \pm 2.2$ |
| **InternAgent (Team et al., 2025)** | Deepseek-R1 | $88.9 \pm 2.2$ | $24.4 \pm 2.2$ | $0.0 \pm 0.0$ | $4.4 \pm 2.2$ | $20.0 \pm 0.0$ | $24.4 \pm 2.2$ |
| **Famou-Agent (Li et al., 2025)** | Gemini-2.5-Pro | $95.6 \pm 2.2$ | $35.6 \pm 2.2$ | $0.0 \pm 0.0$ | $2.2 \pm 2.2$ | $31.1 \pm 2.2$ | $33.3 \pm 0.0$ |
| **Leeroo (Nadaf et al., 2025)** | Gemini-3-Pro-Preview | $40.0 \pm 0.0$ | $40.0 \pm 0.0$ | $4.4 \pm 2.2$ | $15.6 \pm 5.9$ | $20.0 \pm 7.7$ | $40.0 \pm 0.0$ |
| **ML-Master 2.0 (Zhu et al., 2026)** | Deepseek-V3.2-Speciale | $93.3 \pm 3.8$ | $\underline{44.4} \pm 2.2$ | $2.2 \pm 2.2$ | $6.7 \pm 0.0$ | $\underline{33.3} \pm 0.0$ | $\underline{42.2} \pm 2.2$ |
| **Controlled Evaluation in Our Environment** | | | | | | | |
| **AIDE (Jiang et al., 2025)** | Gemini-2.5-Pro | $68.9 \pm 2.2$ | $15.6 \pm 2.2$ | $0.0 \pm 0.0$ | $2.2 \pm 2.2$ | $13.3 \pm 0.0$ | $15.6 \pm 2.2$ |
| | Gemini-3-Pro-Preview | $55.6 \pm 2.2$ | $20.0 \pm 3.8$ | $0.0 \pm 0.0$ | $0.0 \pm 0.0$ | $17.8 \pm 2.2$ | $17.8 \pm 2.2$ |
| **AIRA-dojo (Toledo et al., 2026)** | Gemini-2.5-Pro | $82.2 \pm 4.4$ | $22.2 \pm 2.2$ | $0.0 \pm 0.0$ | $8.9 \pm 4.4$ | $11.1 \pm 4.4$ | $20.0 \pm 0.0$ |
| | Gemini-3-Pro-Preview | $97.8 \pm 2.2$ | $40.0 \pm 3.8$ | $0.0 \pm 0.0$ | $4.4 \pm 2.2$ | $26.7 \pm 6.7$ | $31.1 \pm 4.4$ |
| **MARS (ours)** | Gemini-2.5-Pro | $91.1 \pm 2.2$ | $35.6 \pm 2.2$ | $4.4 \pm 2.2$ | $2.2 \pm 2.2$ | $24.4 \pm 2.2$ | $31.1 \pm 2.2$ |
| | Gemini-3-Pro-Preview | $100.0 \pm 0.0$ | $42.2 \pm 2.2$ | $0.0 \pm 0.0$ | $4.4 \pm 2.2$ | $\underline{33.3} \pm 3.8$ | $37.8 \pm 2.2$ |
| **MARS+ (ours)** | Gemini-3-Pro-Preview | $100.0 \pm 0.0$ | $\mathbf{57.8} \pm 2.2$ | $4.4 \pm 2.2$ | $2.2 \pm 2.2$ | $\mathbf{37.8} \pm 2.2$ | $\mathbf{44.4} \pm 2.2$ |

## F.2. Generalization to Other Models

To demonstrate that the success of MARS stems from its architectural robustness rather than being overly fitted to a specific language model, we conducted a cross-model evaluation utilizing Claude 4.6 Sonnet. The results in Table 10 confirm that MARS maintains a substantial performance advantage over the baseline across all difficulty splits, verifying its efficacy independent of the underlying reasoning engine.

*Table 10.* Model generalizability on MLE-Bench using Claude 4.6 Sonnet. Values represent Any Medal Rate (Mean $\pm$ SEM) over three independent runs, evaluated on a random sample ($n = 10$ per split) of Lite, Medium, and High tasks.

| Method | Lite | Medium | High | All |
|---|---|---|---|---|
| AIRA-dojo | $40.0 \pm 5.8$ | $6.7 \pm 3.3$ | $13.3 \pm 3.3$ | $20.0 \pm 1.9$ |
| MARS (ours) | $\mathbf{80.0} \pm 0.0$ | $\mathbf{43.3} \pm 3.3$ | $\mathbf{43.3} \pm 3.3$ | $\mathbf{55.6} \pm 1.1$ |

## F.3. Ablation Study Results on MLE-Bench Medium and High Splits

Building upon the ablation study conducted on the **MLE-Bench Lite** split (Section 5.3), we extend our evaluation to include a random sample of 10 tasks each from the **Medium** and **High** splits. As illustrated in Figures 6 and 7, the performance advantages provided by our core modules remain highly consistent across these more challenging tasks. Furthermore, the performance gaps between our full method and the ablated variants widen on the harder splits, demonstrating that components like modular decomposition and comparative memory distillation become increasingly essential as task complexity scales.

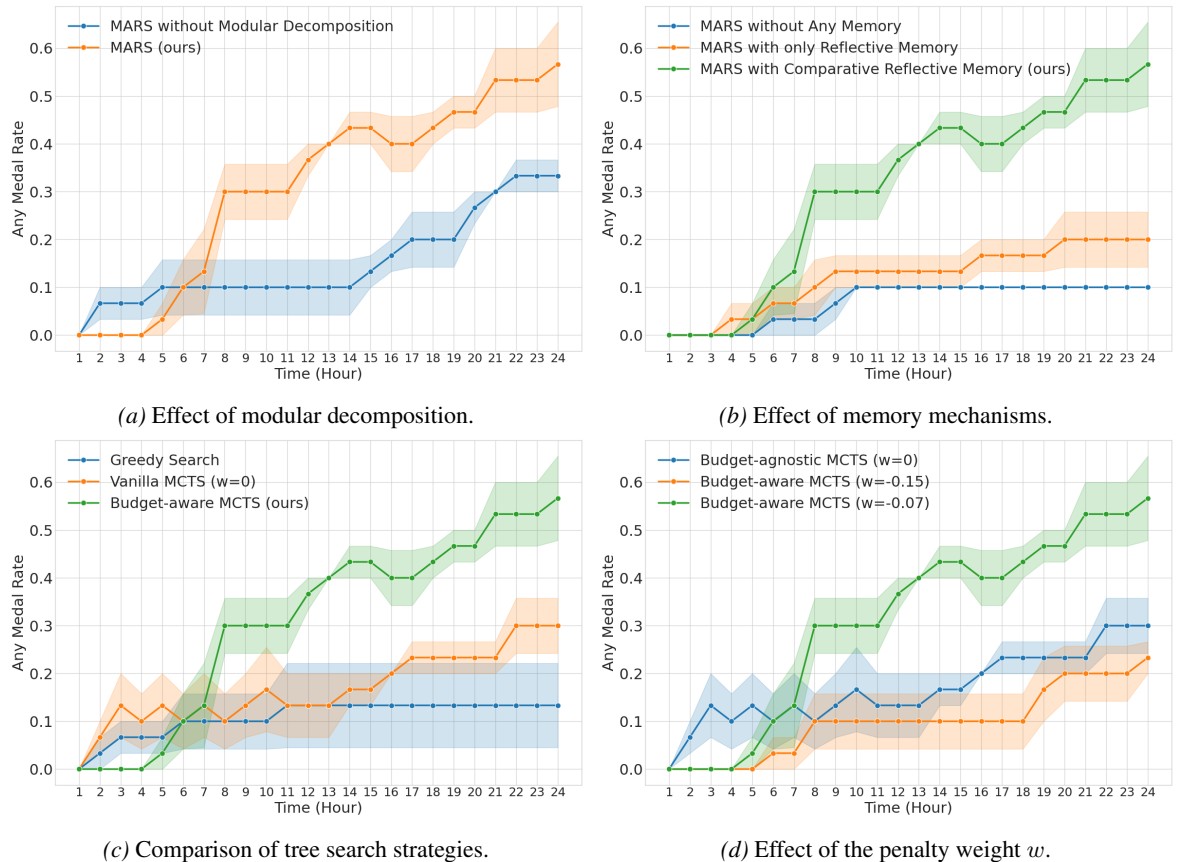

*(a)* Effect of modular decomposition.

*(b)* Effect of memory mechanisms.

*(c)* Comparison of tree search strategies.

*(d)* Effect of the penalty weight $w$.

*Figure 6.* Ablation study results on a random sample of 10 tasks from the MLE-Bench Medium split.

## F.4. Cost-Performance Trade-off

*Table 11.* Cost and performance analysis of different agents using Gemini-2.5-Pro. Metrics are averaged across competitions.

| Metric | AIDE (24 Hour) | AIRA-dojo (24 Hour) | MARS (4 Hour) | MARS (24 Hour) |
|---|---|---|---|---|
| # API Calls | $293.5 \pm 18.9$ | $867.0 \pm 77.0$ | $122.1 \pm 6.0$ | $594.8 \pm 35.4$ |
| # Input Tokens ($\times 10^5$) | $34.6 \pm 4.0$ | $128.3 \pm 13.2$ | $44.3 \pm 8.8$ | $286.6 \pm 54.7$ |
| # Output Tokens ($\times 10^5$) | $6.3 \pm 0.5$ | $23.0 \pm 2.1$ | $1.2 \pm 0.1$ | $6.0 \pm 0.4$ |
| Price ($) | $10.6 \pm 0.9$ | $39.0 \pm 3.7$ | $9.6 \pm 2.2$ | $60.5 \pm 13.8$ |
| Any Medal Rate (%) | $23.1 \pm 0.4$ | $24.4 \pm 1.2$ | $28.4 \pm 1.2$ | $43.1 \pm 1.6$ |

In this section, we analyze the computational costs associated with our framework compared to the baselines (Table 11). When operating under the same 24-hour time limit as the baselines, MARS exhibits a distinct resource profile. It achieves the lowest generation volume, producing fewer output tokens ($6.0 \times 10^5$) than both baselines and making significantly fewer API calls than AIRA-dojo (594.8 vs. 867.0). However, its input consumption is substantial ($286.6 \times 10^5$ tokens) – approximately 2.2 times that of AIRA-dojo. This increase is largely due to maintaining a comprehensive memory context filled with learned lessons and modular structures. Because the Gemini-2.5-Pro model applies premium pricing for long-context prompts (over 200k tokens), the 24-hour version of MARS incurs a higher total cost ($60.5) than AIRA-dojo ($39.0). Crucially, this investment yields substantial returns: the Any Medal Rate increases from 24.4% to 43.1%, justifying the higher expense through superior efficacy.

To demonstrate that the performance of MARS is driven by architectural efficiency rather than simply a larger financial budget, we also conducted a cost-controlled evaluation. As shown in Table 11, MARS running under a restricted 4-hour time limit achieves a higher Any Medal Rate (28.4%) while spending less ($9.6) than both 24-hour baselines (AIDE and

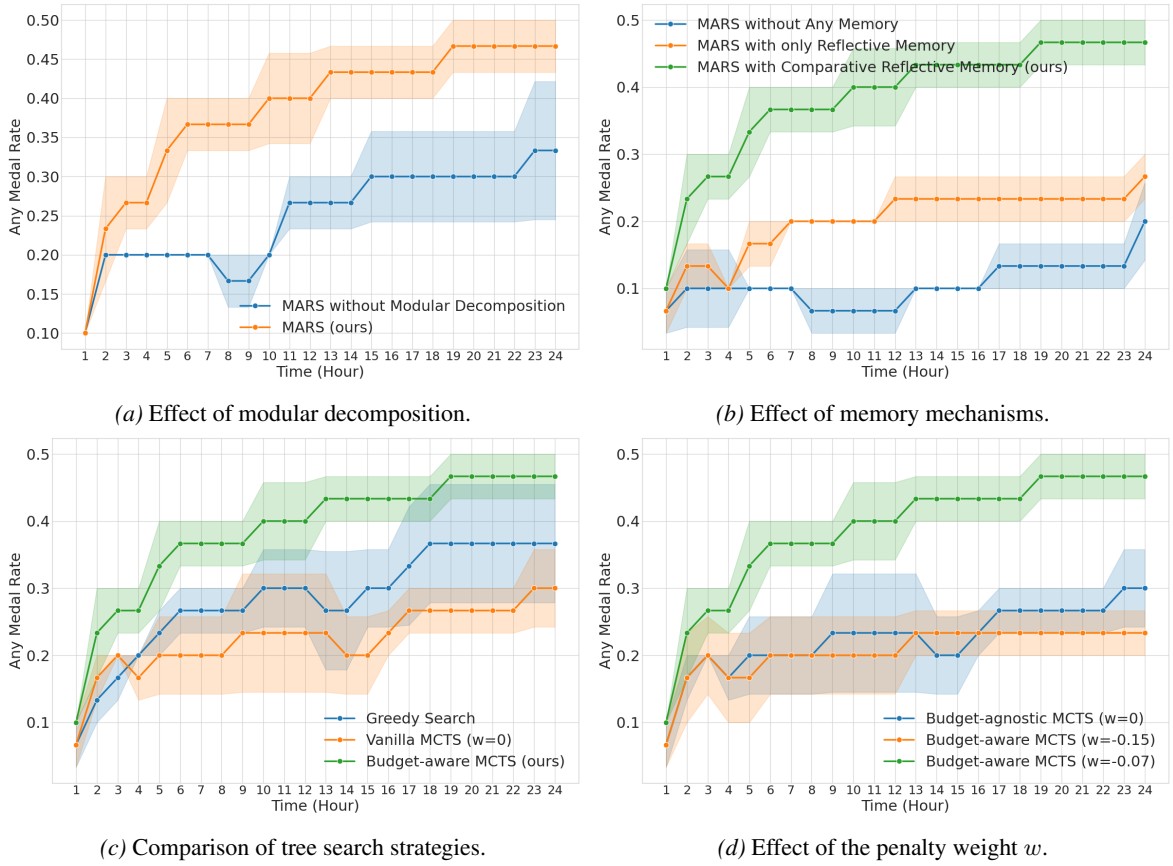

*(a)* Effect of modular decomposition.

*(b)* Effect of memory mechanisms.

*(c)* Comparison of tree search strategies.

*(d)* Effect of the penalty weight $w$.

*Figure 7.* Ablation study results on a random sample of 10 tasks from the MLE-Bench High split.

AIRA-dojo). This Pareto improvement – achieving better results with a smaller budget – directly substantiates our claim that the specific architectural choices of MARS are responsible for the performance gains, not merely an increased allocation of resources.

## G. Qualitative Examples of Causally Correct Distilled Lessons

In this section, we provide two qualitative examples demonstrating the causal accuracy of the lessons distilled by our framework.

**Example 1: Tabular Playground Series (May 2022).** MARS diagnosed a 0.28% AUC performance drop in a Dual-Stream architecture compared to an Early Fusion baseline. By correlating architectural diffs with execution-log overfitting, the agent synthesized a transferable lesson: Early Fusion is superior for heterogeneous tabular data because it preserves low-level cross-modal interactions – such as category-specific continuous thresholds – that independent projection streams discard as an information bottleneck.

**Example 2: Histopathologic Cancer Detection.** MARS diagnosed why a multi-model ensemble underperformed (0.976 AUC) against its own single-model component (0.982 AUC). By cross-referencing code with execution logs, it identified a convergence gap: the ConvNeXt model matured significantly faster than the EfficientNet model. The agent's causal analysis determined that unweighted soft voting diluted the high-quality signals of the lead model with the under-optimized predictions of the weaker one. Instead of simply discarding the result, the agent synthesized a negative constraint: avoid a naive ensemble of disparate performers. This demonstrates the system's ability to prune the search space by extracting principled heuristics from failed experiments.

# H. Prompts

This section provides the full suite of instruction prompts utilized by MARS to orchestrate the various agents involved in solving MLE tasks.

---

**Metric Parsing Instruction**

```
==== Task ====
Your task is to analyze the provided problem description to identify the primary
    evaluation metric and determine if a lower score indicates better performance.
    Your response must be in a specific JSON format with the following fields:
- metric_name (string): This field specifies the name of the primary evaluation
    metric.
- lower_is_better (boolean): This field indicates whether the metric should be
    minimized. If a lower value of the metric represents better performance (e.g.,
    for Mean Squared Error), set this to true. If a higher value represents better
    performance (e.g., for accuracy), set this to false.

# Response Format
Your response should be in the following JSON format in a single markdown code block
    (wrapped in ```):
```json
{{
    "metric_name": "accuracy",
    "lower_is_better": false,
}}
```
```

---

**Metadata Generation Instruction**

```
==== Task ====
Your task is to write a Python script that generates three metadata files for the
    training, validation, and test datasets respectively. This metadata (e.g., sample
    IDs, file paths, labels) will be used by other scripts to load data efficiently.

# Requirements
- The script's only responsibility is to generate metadata. It should not perform
    any model training or inference.
- The script must read raw data from the `./input` directory. This directory should
    be treated as read-only.
- All generated metadata files (e.g., .csv, .parquet, .json) must be saved directly
    to the `./metadata` directory.
- You must not copy or move the original raw data. The `./metadata` directory should
    only contain the newly generated metadata files.
- All file paths stored within the metadata must be relative to the `./input`
    directory. Review the Dataset Information section above to identify the correct
    file paths and structure.
- The metadata for the training and validation datasets must include the ground-
    truth labels.
- Create a validation set by splitting the training data only if a separate
    validation dataset is not already available.
    - Use a fixed 80:20 ratio (80% training and 20% validation). This ratio should
    not be a user-configurable argument.
    - Randomly shuffle the training data before splitting. To ensure the split is
    reproducible, use a fixed random state (`RANDOM_STATE = 42`).
    - Apply stratified sampling or group sampling to ensure the validation dataset's
     distribution properly represents the original data.
        - Stratified Sampling: Use this if it's a classification task, stratifying
    by the target label.
        - Group Sampling: Use this if the data has inherent groups (e.g., patient
    IDs, user IDs) that must not be split across the train and validation sets.
```

– After generating the metadata, the script must immediately load the datasets using
  the new metadata and perform the following checks:
  – Print summary statistics for the final training, validation, and test datasets
   (e.g., total number of samples, class distributions, data shapes, number of
  unique users, etc.).
  – If the metadata contains file paths, programmatically check 1000 relative file
   paths randomly selected from each of the metadata files. Calculate the ratio of
  paths that do not resolve to an existing file in `./input`. If this "missing file
   ratio" is greater than 0.5, the script must raise an error. Before raising the
  error, print a sample (e.g., the first five) of the non-resolving file paths to
  the console for debugging purposes.
  – If a new validation set was created, you must programmatically verify that it
  satisfies the requirements.
      – Assert that the stratification or group split was successful.
      – Raise an error (e.g., AssertionError) if these verification checks fail.

# Implementation Guideline
– The code should be a single-file python program that is self-contained and can be
  executed as-is.
– The script must be complete and able to run from start to finish without premature
   termination or skipped sections.
– Your response should only contain a single code block.
– All validation checks must fail explicitly, either by raising an Exception or
  triggering an AssertionError.
– Do not use `try ... except` blocks to suppress or ignore errors in your code
  examples.
– Be aware of the running time of the code, it should complete within {exec_timeout}.

– All the provided input data is stored in "./input" directory. There is no need to
  unzip any files.

---

## Validation Dataset Verification Instruction

==== Task ====
Analyze the provided Python script and its execution output to verify if the
    validation dataset was handled or created successfully.

# Python Script
{code}

# Execution Output
{term_out}

# Requirements
You must review the script and output based on the criteria below. Your entire
    response must be a single JSON code block.

– Success Criteria: The success field must be set to True if one of the following
    two conditions is met. Otherwise, set it to False.
    1. Existing Validation Set: The script correctly identifies that a separate
    validation dataset is already available in the raw data (i.e., no new split is
    required).
    2. Created Validation Set: The script correctly creates a new validation set by
    splitting the training data. \
Your analysis must confirm that the script's logic properly attempts to create a
    representative split (e.g., by using stratified or group sampling).
– JSON Response Format: Provide your review in the following JSON format.
    – analysis (string): A concise rationale for your decision.
        – If successful: Explain which of the two success criteria was met.
        – If failed: Briefly explain why the script failed to meet either criterion
    (e.g., "The script split the data randomly instead of using stratification.").

```
    – success (bool): True if the validation dataset was handled or created
    successfully, False otherwise.

# Response Format
The review must be submitted in the following JSON format in a single markdown code
    block (wrapped in ```):
```json
{{
    "analysis": "The validation dataset was not created successfully. The script
    split the training data but did not use stratified sampling, failing to create a
    representative sample.",,
    "success": false,
}}
```
```

## Metadata Documentation Instruction

```
==== Task ====
Your task is to analyze the provided Python script and its execution output to
    create clear documentation for each file generated in the `./metadata` directory.

# Python Script
{code}

# Execution Output
{term_out}

# Requirements
For each file generated in the `./metadata` directory, provide a detailed breakdown
    covering:
- Content and Purpose:
    – Describe the information or data contained within the file (e.g., "Contains
    image_id, file_path, and label for the training set.").
    – Explain its primary purpose (e.g., "This file is used by the data loader to
    find image files and match them with their correct labels.").
- Schema / Structure: Detail the structure, such as column names, data types, and an
    example row if applicable.
- Loading Method: Explain the standard method or library function required to load
    this file (e.g., "Load with pandas.read_csv()" or "Load with joblib.load()").
```

## Exploratory Data Analysis Instruction

```
==== Task ====
Your task is to write a robust Python script to perform an Exploratory Data Analysis
    (EDA) on the training dataset. The script must adapt its analysis based on the
    data modality (Tabular, Image, Audio, or Text). The output should act as a report
    to inform feature engineering and preprocessing strategies.

# Requirements
1. Data Integrity: Ensure all analysis is strictly performed on the training set to
    prevent data leakage.
2. Target Variable Analysis
- Distribution: Calculate the distribution of the target variable.
- Imbalance/Skew:
    – If Classification: Calculate class balance ratios.
    – If Regression: Calculate Skewness and Kurtosis to assess normality.
3. Input Data Analysis (Modality-Specific)
- If Tabular Data:
    – Numerical: Report mean, std, min, max, and outlier counts (IQR method).
```

```
    - Categorical: Report cardinality; flag columns with > 50 categories or rare
    labels (< 1 percent frequency).
    - Missing Values: Report count/percentage of NaNs per column.
- If Image Data:
    - Dimensions: Analyze distributions of Image Widths, Heights, and Aspect Ratios.
    - Channels: Report the distribution of channel counts (e.g., Grayscale vs. RGB).
    - Pixel Stats: Calculate the global mean and standard deviation of pixel values
    (for normalization).
- If Audio Data:
    - Signal: Analyze distributions of Duration (seconds), Sampling Rates, and Bit
    Depths.
    - Channels: Check for mono vs. stereo inconsistency.
- If Text Data:
    - Lengths: Analyze distribution of sequence lengths (character and word counts).
    - Vocabulary: Report unique vocabulary size and OOV (Out of Vocabulary)
    potential.
4. Feature/Signal Relationships
- Structured (Tabular) Relationships:
    - Correlation: Pearson/Spearman for numerical; Mutual Information for
    categorical.
    - Importance: Train a lightweight Random Forest and report top 5 features.
    - Redundancy: Report collinear pairs (Correlation > 0.90).
- Unstructured (Meta-Feature) Relationships: Analyze the relationship between
    metadata and the target (e.g., "Do longer audio files correlate with specific
    classes?", "Are larger images associated with higher regression targets?").
5. Formatting & Output
- Organize the output into distinct, capitalized sections.
- Use f-strings to format floats to 4 decimal places for readability.
```

## Model Architecture Search Instruction

```
==== Task ====
Your task is to propose {num_model_candidates} distinct model architectures to solve
    the problem. **Action:** Use Google Search to research state-of-the-art and
    efficient architectures relevant to this domain.

# Requirements
- **Broad Diversity:** The candidates must represent different algorithmic families.
    Do not propose multiple variations of the same underlying method (e.g., do not
    suggest two different ResNets). Aim for a mix of:
    * Instance-Based / Kernel Methods (e.g., k-NN, SVM)
    * Tree-Based Ensembles (e.g., LightGBM, XGBoost, CatBoost)
    * Deep Learning (e.g., CNN, MLP, Transformers, RNNs)
- **Problem Alignment:** Architectures must be specifically tailored to the data
    modality (e.g., tabular, image, time-series) and input structure.
- **Hybridization:** Incorporate hybrid or ensemble designs if they offer a clear
    advantage for heterogeneous data.
- **Efficiency First:** Prioritize "lightweight" designs. For each family, choose
    the architecture that offers the best trade-off between low computational cost (
    fast training/inference) and high performance.
- **Data Constraints:** If the training data is limited, explicitly address
    regularization or low-complexity designs to prevent overfitting.
- For each model, create a JSON object with the following two keys:
    - `reasoning`: Justification for why this architecture fits the constraints (
    efficiency, data size, and why it was chosen over others in its category).
    - `description`: A technical description of the architecture and design
    philosophy.

# Response Format
Your response should be in the following JSON format in a single markdown code block
    (wrapped in ```):
```

```json
[
    {{"reasoning": "k-NN is small and efficient...", "description": "We can use K-NN
     for this task..."}},
    {{"reasoning": "CNN is effective and efficient...", "description": "We can use
    CNN for this task..."}},
    {{"reasoning": "GBMs is an effective model...", "description": "We can use GBMs
    for this task..."}},
]
```

## Initial Idea Proposal Instruction

```
==== Model Architectures ====
{model_arch_desc}

==== Previous Ideas ====
{previous_ideas}

==== Task ====
Your task is to propose a highly efficient **baseline approach** to solve the
    problem.

# Requirements
- Novelty: The proposed solution must remain strictly distinct from the approaches
    listed in Previous Ideas.
- Model Design: Synthesize a simple and lightweight architecture using the provided
    Model Architectures as a conceptual foundation. Ensure the design is unique and
    has not been suggested in the Previous Ideas.
- Philosophy: Prioritize speed and simplicity over maximum accuracy. Exclude
    resource-intensive techniques, such as heavy augmentations or ensembles, to
    establish a reliable performance baseline.

# Response Format
Your solution must be outlined in natural language without using code or specific
    implementation details. Your response should cover the following aspects:
- Model: Describe the model architecture's design and key components.
- Data: Describe the necessary steps to preprocess data for both training and
    evaluation.
- Training: Outline the training procedure, including key techniques (e.g., loss
    functions, optimizers, or training strategies).
- Evaluation: Describe the process for generating predictions on the test data.
```

## Idea Improvement Instruction

```
==== Previous Ideas ====
{previous_ideas}

==== Lessons ====
{lessons}

==== Task ====
Using the insights from the lessons learned during solution development, your task
    is to propose an optimized strategy to solve the problem more effectively. You
    must synthesize the provided "Lessons" to propose a structural evolution of the "
    Previous Ideas".

# Requirements
```

```
- Structural Innovation (Exploration): Do not propose trivial hyperparameter tuning.
    You must introduce a fundamental change (e.g., a new backbone architecture, a
   multi-stage pipeline, or a distinct feature engineering paradigm) to address
   identified weaknesses.
- Strategic Retention (Exploitation): Explicitly preserve components identified as
    successful in the "Lessons". Do not discard what is already working.
- Computational Budget: The solution is allowed to be moderately heavier than
    previous ideas (e.g., using a stronger backbone), but it must remain feasible for
     standard training environments.
- Citation: Whenever you apply a specific concept or solution from these lessons,
    you must immediately reference it by appending "Cite {{lesson_id}}" to the
    relevant statement.

# Response Format
Your solution must be outlined in natural language without using code or specific
    implementation details. Your response should cover the following aspects:
- Model: Describe the model architecture's design and key components.
- Data: Describe the necessary steps to preprocess data for both training and
    evaluation.
- Training: Outline the training procedure, including key techniques (e.g., loss
    functions, optimizers, or training strategies).
- Evaluation: Describe the process for generating predictions on the test data.
```

## Modular Decomposition Instruction

```
==== Idea ====
{idea}

==== Task ====
Your task is to design a modular repository structure to implement the given idea.
    Do not generate the full code yet; focus on the natural description of the **
    architectural logic**.

# Requirements
- **Modularity:** Break the solution into logical modules based on functionality (e.
    g., data handling, core training and evaluation logic, utilities).
- **Entry Point:** You must include a `main` module that acts as the entry point to
    execute the end-to-end pipeline.
- **Detail:** For each module, the description must include:
    - The purpose of the module.
    - The names of specific classes or functions to be implemented.
    - A brief description of the implementation logic.
    - A brief explanation of how this module interacts with others.
- **Ordering:** The JSON output must be ordered topologically (dependencies first,
    dependent modules last).

# Response Format
Provide the output strictly as a JSON object in a single markdown code block. The
    keys should be the module names and the values should be the detailed
    descriptions. The module name must not include the `.py` extension.

Example Format:
```json
{{
    "module_name": "Implements [Specific Class] to handle [Specific Task]. includes
    functions like [func_a] and [func_b].",
    "main": "Orchestrates the workflow. Imports DataLoader from the data module and
    Model from the model module to run the pipeline."
}}
```
```

## Module Implementation Instruction

```
==== Idea ====
{idea}

==== Python Files ====
The following Python files are already provided. Do not modify them.
{library_files}

==== Target File Description ({file_name}) ====
{file_description}

==== Task ====
Your task is to implement the `{file_name}` module based on the description above.

# Requirements
- Import the functions or classes from the given Python files instead of re-
   implementing them.
- Only implement the module class/functions. DO NOT include an if `__name__ == "
   __main__":` block. DO NOT implement the end-to-end pipeline.
- Ensure functions accept arguments for flexibility. You must include
   hyperparameters to control dataset size (for debugging) and training steps/epochs
   .
- When printing validation metrics, please print the full precision without any
   rounding or formatting.
- If loading raw data, use the metadata in `./metadata` to identify the correct train
   /val/test splits.
- If this module performs deterministic data processing, you must implement a
   caching mechanism strictly following this logic:
   - **Function Signature:** The processing function must accept a `
   load_cached_data: bool` argument.
   - **Directory Safety:** Ensure the directory `./working/{dir_name}/` exists (use
   `os.makedirs(..., exist_ok=True)`).
   - **Prohibited:** Do NOT use `pickle`. Use `parquet` (via pandas) or `npy` (via
   numpy).
   - **Logic Flow:**
        1. IF `load_cached_data` is True: Try to load the file.
        2. IF loading fails (file missing or corrupt) OR `load_cached_data` is False:
            - Compute/process the data from scratch.
            - Save the result to the cache directory `./working/{dir_name}/` for
   future runs.
        3. Return the data.
- If this module handles model training:
   - **Metrics:** Print key training and validation metrics during training process.

   - **Optimization:** Implement Early Stopping to prevent overfitting and reduce
   runtime.
- If this module handles submission generation:
   - Generate predictions for the entire test set. Save the final predictions to `./
   submission/submission.csv`.
   - Refer to the sample submission file (e.g., `./input/sample_submission.csv` or
   `./input/sampleSubmission.csv`) for the correct formatting required by the
   competition.
```

## Module Testing Instruction

```
==== Python Files ====
The following Python files are already provided. Do not modify them.
{library_files}
```

```
==== Task ====
Your task is to write code examples demonstrating how to instantiate and utilize the
    classes or functions from the provided Python files.

# Requirements
- Optimize for Speed: Limit hyperparameters (e.g., reduce the number of epochs/steps,
    use a smaller dataset subset) to ensure the demonstration executes quickly.
- Verify Logic: Include assertions or validation steps to confirm the correctness of
    complex classes and functions. \
You may skip verification for trivial components, such as configuration classes.
```

## Solution Drafting Instruction

```
==== Idea ====
{idea}

==== Python Files ====
The following Python files are already provided. Do not modify them.
{library_files}

==== Target File Description (`runfile.py`) ====
{file_description}

==== Task ====
Your task is to implement the end-to-end orchestration script `runfile.py`. This
   script serves as a **fast baseline** to verify the idea. It must train the model,
    validate performance, perform failure analysis, and generate a submission.

# Requirements
- Import the functions or classes from the given Python files instead of re-
   implementing them.
- Make the model training fast.
    - Limit maximum number of training samples and training steps/epochs to ensure a
     quick baseline execution.
    - Set appropriate batch sizes to prevent CUDA out-of-memory errors.
- After training is complete, you must execute validation assessment, failure
   analysis, and submission generation.
    - You must load the hold-out validation dataset using the metadata located in
    the `./metadata` directory.
    - You must print the final validation metric computed on the entire hold-out
    validation set in this format `Final Validation Metric: <value>`. Without this
    metric, the solution cannot be evaluated, rendering the entire code invalid. You
    must use the validation metric defined in the Task Description. Please print the
    full precision of the validation metric without any rounding or formatting.
    - You must perform failure analysis on the trained model. You must perform
    failure analysis on the validation set to identify systematic error patterns.
    Calculate and print the correlation between the model's error magnitude and the
    input features to reveal which variables are most associated with poor
    performance.
    - You must generate predictions for the entire test set and create the
    submission file{submission_cond}. Save the final predictions to `./submission/
    submission.csv`. Refer to the sample submission file (e.g., `./input/
    sample_submission.csv` or `./input/sampleSubmission.csv`) for the correct
    formatting required by the competition.
- Optimize the validation inference speed.
    - Ensure the model is in evaluation mode for this inference.
    - Your code must automatically detect and utilize an available GPU for inference.
     Ensure the model and all data batches are moved to the correct device (GPU or
    CPU).
    - During inference, you don't need to compute gradients. Disabling this process
    reduces memory consumption and speeds up computation.
```

```
– Call data loading functions with `load_cached_data=True` (if applicable) to utilize
     preprocessed data in the `./working` directory.
```

## Solution Improvement Instruction

```
==== Lessons ====
{lessons}

==== Previous Solution ====
{previous_solution}

==== Task ====
Your task is to modify the Python files from the previous solution to optimize
    performance.

# Requirements
- Modifications must be targeted and specific (ablation-style). Do not rewrite the
    entire solution; focus on isolating and improving specific aspects.
- You should apply the relevant knowledge provided in the Lessons section to support
     your optimization strategy. Whenever you apply a specific concept or solution
    from these lessons, you must immediately reference it by appending "Cite {{
    lesson_id}}" to the relevant statement.
- Optimize hyperparameter settings (e.g., training steps, learning rate, batch size)
     to strike the best balance between predictive performance and execution speed.
- **Do not remove** the following core logic from the original `runfile.py` script:
     – Print the final validation metric computed on the entire hold-out validation
    set.
     – Perform failure analysis on the trained model.
     – Generate predictions for the entire test set and create the submission file{
    submission_cond}.
```

## Bug Analysis Instruction

```
==== Debug Lessons ====
{lessons}

==== Python Files ====
The following Python files are already provided.
{files}

==== Task ====
You are tasked with debugging a script failure. You should summarize the execution
    traceback and explain the root cause of the errors. You should apply the relevant
     knowledge provided in the Debug Lessons section to support your diagnosis.
    Whenever you apply a specific concept or solution from these lessons, you must
    immediately reference it by appending "Cite {{lesson_id}}" to the relevant
    statement. You can use Google Search as needed for debugging.

Execution Traceback (`python runfile.py`):
{exec_result}
```

## Debugging Instruction

```
==== Debug Lessons ====
{lessons}
```

```
==== Python Files ====
The following Python files are already provided. Do not modify them.
{files}

==== Task ====
We ran this command (`python runfile.py`) and got some errors.

Execution Traceback:
{exec_result}

Error analysis:
{error_analysis}

Your task is to revise the given Python files to fix the errors based on the
    provided error analysis. You can use Google Search as needed for debugging.

# Requirements
- You should write a brief natural language description of what the issue in the
    previous implementation is and how the issue can be fixed.
- The fix must be targeted. Do not change the core logic or intended functionality
    of the original code; only correct the specific implementation error shown in the
     Execution Traceback.
- You should apply the relevant knowledge provided in the Debug Lessons section to
    guide your fixes. Whenever you apply a specific concept or solution from these
    lessons, you must immediately reference it by appending "Cite {{lesson_id}}" to
    the relevant statement.
- Do not use `try...except` blocks to catch, suppress, or ignore the original error.
    The fix must address the root cause of the problem.
```

## Debugging Lesson Distillation Instruction

```
You are an expert Python debugger and instructor. Your task is to analyze a
    debugging attempt and distill a high-value "Lesson Learned".

# Input
Initial State:
{source_files}

Initial Execution Traceback:
{source_exec_result}

Initial Error analysis:
{source_error_analysis}

Attempted Fix (Diff):
{diff}

Execution Traceback after applying the fix:
{final_exec_result}

# Guidelines
- Determine if the Attempted Fix resolved the Initial Error based on the Result of
    Fix.
- If the fix SUCCEEDED: Explain the root cause of the initial error and why this
    specific fix was the correct solution.
- If the fix FAILED: Explain why the attempted fix was insufficient, incorrect, or
    introduced new issues. The lesson must focus on avoiding this specific pitfall.

# Response Format
- Title: A concise, imperative, and memorable summary of the lesson.
```

- Explanation: A clear paragraph synthesizing the error context. Describe the
  specific mechanism of the failure and the logic required to fix it.
- Detection: How to identify this issue in the future. List specific signals, such
  as particular Exception types, stack trace patterns, or code smells.

---

**Execution Result Review Instruction**

```
==== Python Files ====
The following Python files are already provided.
{library_files}

==== Task ====
Your task is to evaluate the output of the code execution for the provided code and
    report the empirical findings. The review must be submitted in a specific JSON
    format with the following fields:
- summary (string): In this field, provide a brief summary describing the empirical
    findings. This must include:
    - The training loss trend (e.g., did it converge/minimize?).
    - Failure analysis.
    - The final validation metric.
    - The reasoning for your `valid_metric` assessment (e.g., "The final validation
    metric is valid," or "The final validation metric is invalid due to validation
    data leakage...").
- metric (number or null): Report the value of the validation metric here. You must
    convert percentages to decimals (e.g., 95% -> 0.95). This should be null if the
    metric cannot be found or determined.
- valid_metric (boolean): Set to `true` if the final validation metric is valid. Set
    to `false` if any of the following conditions are met:
    - The computed final validation metric does not match the one defined in the
    Task Description.
    - The final validation metric is calculated incorrectly.
    - The final validation metric is not computed on the entire hold-out validation
    set.
    - There are signs of validation data leakage (e.g., the validation set was used
    in training).

Code:
```
{code}
```

Execution Output:
{term_out}

# Response Format
The review must be submitted in the following JSON format in a single markdown code
    block (wrapped in ```):
```json
{{
    "summary": "The code trains a model to solve the task... The final validation
    metric is ...",
    "metric": 0.99,
    "valid_metric": true,
}}
```
```

---

**Solution Lesson Distillation Instruction**

```
==== Current Best Solution ====
{best_solution}

==== New Solution ====
{new_solution}

==== Task ====
Your task is to analyze the provided solutions to distill a high-value "Lesson
    Learned".

# Guidelines
- **Check Context:**
   - If *Current Best Solution* exists: Comparative Analysis. Contrast the
     algorithmic approach of the New vs. Current. Explain precisely *why* the New
     Solution improves or degrades performance based on the execution results.
   - If *Current Best Solution* is missing: Empirical Analysis. Summarize the
     findings and effectiveness of the New Solution based on its execution results.
- **Logic over Syntax:** Focus on algorithmic choices, data structures, and
    architectural decisions. Ignore minor syntactic sugar unless it affects
    performance.
- **Causal Chain:** Trace the logic to prove exactly how the new approach resolves
    the specific bottleneck.
- **Generalizability:** The final lesson must be abstract enough to apply to similar
     problems in the future, not just this specific snippet.

# Response Format
- Title: A clear, memorable title for the lesson.
- Summary: A brief, high-level overview of the methods or algorithmic changes
    applied in the New Solution.
- Empirical Findings: Analysis of the execution results. If comparing, highlight the
     delta in performance (validation metric and execution time) and the specific
    trade-offs observed.
- Key Lesson: A standalone, actionable principle. Write this as a heuristic or rule
    of thumb (e.g., "When handling sparse matrices, prefer X over Y because..."). If
    a developer reads *only* this paragraph, they should learn a technique to apply
    in their own work.
```

---

**Lesson Deduplication Instruction**

```
You are a Machine Learning Engineer responsible for maintaining a knowledge base of
    technical lessons.

==== Existing Lessons ====
{existing_lessons}

==== New Lesson ====
{new_lesson}

==== Task ====
Your task is to determine if the **New Lesson** is semantically equivalent to any of
     the **Existing Lessons**.

### Guidelines
- **Semantic Overlap:** A lesson is a duplicate if the core insight, principle, or
    actionable advice is effectively the same, even if the wording differs.
- **Subsets:** If the **New Lesson** is fully covered by a broader existing lesson,
    count it as a duplicate.
- **Novelty:** If the **New Lesson** provides a specific nuance, edge case, or
    context not covered by existing lessons, it is **not** a duplicate.
```

```
# Response Format
Provide your analysis in a single valid JSON object inside a single markdown code
    block (wrapped in ```).

**Fields:**
- `reasoning` (string): Briefly explain your decision. If a duplicate exists,
    explicitly quote or summarize the specific existing lesson that overlaps.
- `duplicate` (boolean): Use `true` if it is a duplicate, `false` otherwise.

**Example Output:**
```json
{{
    "reasoning": "The lesson is the same as one of the existing lessons.",
    "duplicate": true
}}
```
```

# I. Example Solution Code

In this section, we showcase the final solution generated for the iMet-2020-fgvc7 competition. This repository was produced by MARS utilizing Gemini-3-Pro-Preview and achieved the highest validation performance during the search process.

## Module: library/config.py

```python
import os
import torch
import random
import numpy as np

class Config:
    # =========================================================================
    # Directories & Paths
    # =========================================================================
    INPUT_DIR = "./input"
    METADATA_DIR = "./metadata"

    # Output directories
    WORKING_DIR = "./working/idea_4"
    SUBMISSION_DIR = "./submission"

    # Metadata Files (Generated by previous steps)
    TRAIN_CSV = os.path.join(METADATA_DIR, "train.csv")
    VAL_CSV = os.path.join(METADATA_DIR, "val.csv")
    TEST_CSV = os.path.join(METADATA_DIR, "test.csv")

    # Raw Data Files
    LABELS_CSV = os.path.join(INPUT_DIR, "labels.csv")
    SAMPLE_SUBMISSION = os.path.join(INPUT_DIR, "sample_submission.csv")

    # =========================================================================
    # Data Configuration
    # =========================================================================
    IMG_SIZE = (352, 352)  # Resolution optimized for detail vs throughput
    NUM_CLASSES = 3474  # Total number of attributes

    # =========================================================================
    # Model Configuration
    # =========================================================================
    # Ensemble members
    MODEL_A_NAME = "resnet101d"  # Texture specialist (Deep Stem)
    MODEL_B_NAME = "convnext_base"  # Context specialist (Large Kernel)

    # =========================================================================
    # Training Hyperparameters
    # =========================================================================
    SEED = 42
    EPOCHS = 12
    BATCH_SIZE = 48  # Tuned for A100 40GB with 352x352 resolution

    # Optimizer settings
    LEARNING_RATE = 1e-3  # Max LR for OneCycle
    WEIGHT_DECAY = 1e-2  # Standard AdamW decay

    # Loss settings
    ASL_GAMMA_NEG = 4.0  # Asymmetric Loss parameters
    ASL_GAMMA_POS = 0.0
    ASL_CLIP = 0.05

    # =========================================================================
```

```
    # Compute Environment
    # ==========================================================================
    NUM_WORKERS = 8  # 12 vCPUs available, leave some overhead
    DEVICE = "cuda" if torch.cuda.is_available() else "cpu"

    # ==========================================================================
    # Debugging & Development
    # ==========================================================================
    DEBUG = False  # Set to True to run on a small subset
    DEBUG_SAMPLE_SIZE = 5000

    @classmethod
    def setup(cls):
        """Creates necessary output directories."""
        os.makedirs(cls.WORKING_DIR, exist_ok=True)
        os.makedirs(cls.SUBMISSION_DIR, exist_ok=True)

def seed_everything(seed=42):
    """Sets the random seed for reproducibility across all libraries."""
    random.seed(seed)
    os.environ["PYTHONHASHSEED"] = str(seed)
    np.random.seed(seed)
    torch.manual_seed(seed)
    torch.cuda.manual_seed(seed)
    torch.backends.cudnn.deterministic = True
    torch.backends.cudnn.benchmark = False  # False for exact reproducibility

# Initialize directories on import
Config.setup()
```

### Module: library/dataset.py

```
import os
import cv2
import torch
import numpy as np
import pandas as pd
import albumentations as A
from albumentations.pytorch import ToTensorV2
from torch.utils.data import Dataset, DataLoader
from library.config import Config

def load_processed_dataframe(mode, load_cached_data=True):
    """
    Loads the metadata dataframe for a specific mode (train/val/test).
    Implements caching using Parquet files to store processed dataframes
    (where attribute_ids strings are converted to lists).
    """
    cache_dir = Config.WORKING_DIR
    os.makedirs(cache_dir, exist_ok=True)
    cache_path = os.path.join(cache_dir, f"cached_{mode}.parquet")

    # 1. Try to load from cache
    if load_cached_data and os.path.exists(cache_path):
        try:
            df = pd.read_parquet(cache_path)
            return df
        except Exception:
```

```
                # If load fails, proceed to process from scratch
                pass

        # 2. Process from scratch
        if mode == "train":
            csv_path = Config.TRAIN_CSV
        elif mode == "val":
            csv_path = Config.VAL_CSV
        elif mode == "test":
            csv_path = Config.TEST_CSV
        else:
            raise ValueError(f"Unknown mode: {mode}")

        if not os.path.exists(csv_path):
            raise FileNotFoundError(f"Metadata file not found: {csv_path}")

        df = pd.read_csv(csv_path)

        # Parse attribute_ids: "0 1 2" -> [0, 1, 2]
        # Handle NaNs by converting to empty string first
        df["attribute_ids"] = df["attribute_ids"].fillna("")

        # Function to safe convert string to list of ints
        def parse_ids(x):
            if not x.strip():
                return np.array([], dtype=int)
            return np.array([int(i) for i in x.split()], dtype=int)

        # We store as numpy arrays inside the dataframe cells for parquet compatibility/
        efficiency
        df["parsed_attributes"] = df["attribute_ids"].apply(parse_ids)

        # 3. Save to cache
        try:
            df.to_parquet(cache_path, index=False)
        except Exception as e:
            print(f"Warning: Failed to save cache to {cache_path}: {e}")

        return df

class ArtworkDataset(Dataset):
    def __init__(self, df, transforms=None, mode="train"):
        """
        Args:
            df (pd.DataFrame): DataFrame containing metadata.
            transforms (albumentations.Compose): Transforms to apply.
            mode (str): 'train', 'val', or 'test'.
        """
        self.df = df
        self.transforms = transforms
        self.mode = mode
        self.input_dir = Config.INPUT_DIR
        self.num_classes = Config.NUM_CLASSES

    def __len__(self):
        return len(self.df)

    def __getitem__(self, idx):
        row = self.df.iloc[idx]
        image_id = row["id"]
        file_path = row["file_path"]
```

```python
        # Load Image
        full_path = os.path.join(self.input_dir, file_path)
        image = cv2.imread(full_path)

        if image is None:
            # Fallback for missing/corrupt images: return black image
            # This prevents crashing during training
            image = np.zeros(
                (Config.IMG_SIZE[0], Config.IMG_SIZE[1], 3), dtype=np.uint8
            )
        else:
            image = cv2.cvtColor(image, cv2.COLOR_BGR2RGB)

        # Apply Transforms
        if self.transforms:
            augmented = self.transforms(image=image)
            image = augmented["image"]

        # Create Target (Multi-hot encoding)
        target = torch.zeros(self.num_classes, dtype=torch.float32)

        # In test mode, we might not have valid labels, but we still return a dummy
    target
        # parsed_attributes is a numpy array of integers
        attr_ids = row["parsed_attributes"]

        if len(attr_ids) > 0:
            # Ensure indices are within bounds
            valid_ids = attr_ids[attr_ids < self.num_classes]
            target[valid_ids] = 1.0

        return image, target, image_id

def get_transforms(mode="train"):
    """
    Returns the Albumentations transform pipeline for the specified mode.
    """
    img_size = Config.IMG_SIZE

    if mode == "train":
        return A.Compose(
            [
                A.Resize(height=img_size[0], width=img_size[1]),
                A.HorizontalFlip(p=0.5),
                A.Normalize(
                    mean=(0.485, 0.456, 0.406),
                    std=(0.229, 0.224, 0.225),
                    max_pixel_value=255.0,
                    p=1.0,
                ),
                ToTensorV2(),
            ]
        )
    else:
        # Validation and Test
        return A.Compose(
            [
                A.Resize(height=img_size[0], width=img_size[1]),
                A.Normalize(
                    mean=(0.485, 0.456, 0.406),
                    std=(0.229, 0.224, 0.225),
                    max_pixel_value=255.0,
```

```
                p=1.0,
            ),
            ToTensorV2(),
        ]
    )

def get_dataloaders(
    debug=False,
    batch_size=Config.BATCH_SIZE,
    num_workers=Config.NUM_WORKERS,
    load_cached_data=True,
):
    """
    Creates and returns DataLoaders for train, validation, and test sets.

    Args:
        debug (bool): If True, subsets the data for quick debugging.
        batch_size (int): Batch size for the dataloaders.
        num_workers (int): Number of worker processes.
        load_cached_data (bool): Whether to use cached dataframes.

    Returns:
        tuple: (train_loader, val_loader, test_loader)
    """
    # Load DataFrames
    train_df = load_processed_dataframe("train", load_cached_data)
    val_df = load_processed_dataframe("val", load_cached_data)
    test_df = load_processed_dataframe("test", load_cached_data)

    # Debug Subsampling
    if debug:
        debug_size = Config.DEBUG_SAMPLE_SIZE
        train_df = train_df.iloc[:debug_size]
        val_df = val_df.iloc[:debug_size]
        test_df = test_df.iloc[:debug_size]

    # Create Datasets
    train_dataset = ArtworkDataset(
        train_df, transforms=get_transforms("train"), mode="train"
    )
    val_dataset = ArtworkDataset(val_df, transforms=get_transforms("val"), mode="val
")
    test_dataset = ArtworkDataset(
        test_df, transforms=get_transforms("test"), mode="test"
    )

    # Create DataLoaders
    train_loader = DataLoader(
        train_dataset,
        batch_size=batch_size,
        shuffle=True,
        num_workers=num_workers,
        pin_memory=True,
        drop_last=True,
    )

    val_loader = DataLoader(
        val_dataset,
        batch_size=batch_size,
        shuffle=False,
        num_workers=num_workers,
        pin_memory=True,
```

```
        drop_last=False,
    )

    test_loader = DataLoader(
        test_dataset,
        batch_size=batch_size,
        shuffle=False,
        num_workers=num_workers,
        pin_memory=True,
        drop_last=False,
    )

    return train_loader, val_loader, test_loader
```

## Module: library/inference.py

```python
import os
import torch
import numpy as np
import pandas as pd
from tqdm import tqdm

from library.config import Config
from library.dataset import get_dataloaders
from library.models import get_model
from library.utils import optimize_threshold, load_checkpoint

def predict_with_tta(model, dataloader, device, mode="val"):
    """
    Generates predictions using Test Time Augmentation (Horizontal Flip).

    Args:
        model (nn.Module): The trained model.
        dataloader (DataLoader): DataLoader for validation or test set.
        device (torch.device): Device to run inference on.
        mode (str): 'val' or 'test'.

    Returns:
        tuple: (all_probs, all_targets_or_ids)
            - all_probs: np.ndarray of shape (N, num_classes)
            - all_targets_or_ids: np.ndarray of targets (if val) or list of ids (if
    test)
    """
    model.eval()
    all_probs = []
    all_targets_or_ids = []

    with torch.no_grad():
        for batch in dataloader:
            images, targets, ids = batch
            images = images.to(device)

            # 1. Forward pass original
            out_orig = model(images)
            probs_orig = torch.sigmoid(out_orig)

            # 2. Forward pass flipped (TTA)
            # Flip along width dimension (dim 3 for NCHW)
            images_flipped = torch.flip(images, dims=[3])
            out_flipped = model(images_flipped)
```

```python
            probs_flipped = torch.sigmoid(out_flipped)

            # 3. Average probabilities
            avg_probs = (probs_orig + probs_flipped) / 2.0

            all_probs.append(avg_probs.cpu().numpy())

            # Collect targets or IDs based on mode
            if mode == "test":
                all_targets_or_ids.extend(ids)
            else:
                all_targets_or_ids.append(targets.numpy())

    all_probs = np.concatenate(all_probs, axis=0)

    if mode != "test":
        all_targets_or_ids = np.concatenate(all_targets_or_ids, axis=0)

    return all_probs, all_targets_or_ids

def get_model_predictions(model_name, mode, dataloader, device, load_cached_data=
    True):
    """
    Gets predictions for a specific model and mode (val/test).
    Implements caching of the raw probability arrays.

    Args:
        model_name (str): Name of the model architecture.
        mode (str): 'val' or 'test'.
        dataloader (DataLoader): The data loader.
        device (torch.device): Compute device.
        load_cached_data (bool): Whether to use cached .npy files.

    Returns:
        tuple: (probs, targets_or_ids)
    """
    cache_dir = Config.WORKING_DIR
    os.makedirs(cache_dir, exist_ok=True)

    probs_path = os.path.join(cache_dir, f"{model_name}_{mode}_probs.npy")
    meta_path = os.path.join(
        cache_dir, f"{model_name}_{mode}_meta.npy"
    )  # targets or ids

    # Try to load from cache
    if load_cached_data and os.path.exists(probs_path) and os.path.exists(meta_path):

        print(f"Loading cached predictions for {model_name} ({mode})...")
        try:
            probs = np.load(probs_path)
            meta = np.load(meta_path, allow_pickle=True)

            if len(probs) != len(dataloader.dataset):
                raise ValueError(
                    f"Cache size mismatch: {len(probs)} vs {len(dataloader.dataset)}"

                )

            return probs, meta
        except Exception as e:
            print(f"Failed to load cache: {e}. Re-running inference.")
```

```python
    # Run inference
    print(f"Running inference for {model_name} ({mode})...")

    # Load model and checkpoint
    model = get_model(model_name, num_classes=Config.NUM_CLASSES, pretrained=False)
    model = model.to(device)

    checkpoint_filename = f"{model_name}_best.pth"
    try:
        load_checkpoint(model, checkpoint_filename, device=device)
    except FileNotFoundError:
        print(
            f"Warning: Checkpoint {checkpoint_filename} not found. Using random
    weights (for debugging only)."
        )

    probs, meta = predict_with_tta(model, dataloader, device, mode=mode)

    # Save to cache
    np.save(probs_path, probs)
    np.save(meta_path, meta)

    # Clean up
    del model
    torch.cuda.empty_cache()

    return probs, meta

def ensemble_predictions(probs_list):
    """
    Averages a list of probability arrays.
    """
    if not probs_list:
        return None
    return np.mean(probs_list, axis=0)

def generate_submission(debug=Config.DEBUG, load_cached_data=True):
    """
    Main function to generate the submission file.

    1. Loads Val and Test loaders.
    2. Gets predictions for Model A and Model B (with TTA).
    3. Ensembles predictions.
    4. Optimizes threshold on Validation set.
    5. Applies threshold to Test set.
    6. Saves submission.csv.
    """
    device = torch.device(Config.DEVICE)

    # 1. Get DataLoaders
    # We don't need the train loader here
    _, val_loader, test_loader = get_dataloaders(
        debug=debug,
        batch_size=Config.BATCH_SIZE,
        num_workers=Config.NUM_WORKERS,
        load_cached_data=load_cached_data,
    )

    models = [Config.MODEL_A_NAME, Config.MODEL_B_NAME]

    # 2. Validation Inference (for Threshold Calibration)
```

```
print("--- Processing Validation Set ---")
val_probs_list = []
val_targets = None

for model_name in models:
    probs, targets = get_model_predictions(
        model_name, "val", val_loader, device, load_cached_data
    )
    val_probs_list.append(probs)
    val_targets = targets  # Targets should be same for all models

# Ensemble Validation
val_ensemble_probs = ensemble_predictions(val_probs_list)

# Optimize Threshold
print("Optimizing threshold on ensemble...")
best_threshold, best_score = optimize_threshold(val_targets, val_ensemble_probs)
print(f"Optimal Threshold: {best_threshold}")
print(f"Validation Micro-F1 Score with Optimal Threshold: {best_score}")

# 3. Test Inference
print("\n--- Processing Test Set ---")
test_probs_list = []
test_ids = None

for model_name in models:
    probs, ids = get_model_predictions(
        model_name, "test", test_loader, device, load_cached_data
    )
    test_probs_list.append(probs)
    test_ids = ids  # IDs should be same for all models

# Ensemble Test
test_ensemble_probs = ensemble_predictions(test_probs_list)

# 4. Generate Submission CSV
print(f"Generating submission with threshold {best_threshold}...")

# Binarize predictions
predictions_bin = (test_ensemble_probs > best_threshold).astype(int)

submission_rows = []
for idx, image_id in enumerate(test_ids):
    # Get indices of positive classes
    pred_indices = np.where(predictions_bin[idx] == 1)[0]

    # Format as space-separated string
    pred_str = " ".join(map(str, pred_indices))

    submission_rows.append({"id": image_id, "attribute_ids": pred_str})

submission_df = pd.DataFrame(submission_rows)

# Save
out_path = os.path.join(Config.SUBMISSION_DIR, "submission.csv")
submission_df.to_csv(out_path, index=False)
print(f"Submission saved to {out_path}")

return best_score
```

**Module: library/loss.py**

```python
import torch
import torch.nn as nn
from library.config import Config

class AsymmetricLoss(nn.Module):
    """
    Asymmetric Loss (ASL) for multi-label classification.

    ASL optimizes the trade-off between precision and recall by decoupling the
    loss components for positive and negative samples. It down-weights easy
    negatives (which are dominant in multi-label settings) to focus learning
    on hard negatives and positive samples.

    Reference: "Asymmetric Loss For Multi-Label Classification" (ICCV 2021)
    """

    def __init__(
        self,
        gamma_neg=Config.ASL_GAMMA_NEG,
        gamma_pos=Config.ASL_GAMMA_POS,
        clip=Config.ASL_CLIP,
        eps=1e-8,
        reduction="mean",
    ):
        """
        Args:
            gamma_neg (float): Focusing parameter for negative samples (down-weights
         easy negatives).
            gamma_pos (float): Focusing parameter for positive samples.
            clip (float): Probability margin for shifting negative samples (hard
    thresholding).
            eps (float): Small constant for numerical stability in logarithms.
            reduction (str): Specifies the reduction to apply to the output: 'none'
    | 'mean' | 'sum'.
        """
        super(AsymmetricLoss, self).__init__()
        self.gamma_neg = gamma_neg
        self.gamma_pos = gamma_pos
        self.clip = clip
        self.eps = eps
        self.reduction = reduction

    def forward(self, x, y):
        """
        Args:
            x (torch.Tensor): Logits (before sigmoid) of shape (N, C).
            y (torch.Tensor): Ground truth labels of shape (N, C) (0 or 1).

        Returns:
            torch.Tensor: Calculated loss.
        """
        # Explicit casting to float32 is crucial to prevent NaN during mixed-
    precision training
        # Logits and targets must be in float32 for stable log/pow computations
        x = x.float()
        y = y.float()

        # Calculate probabilities
        xs_pos = torch.sigmoid(x)
```

```
        # --- Positive Component ---
        # Standard Focal Loss term for positives: -y * (1-p)^gamma_pos * log(p)
        # We clamp the input to log to avoid log(0)
        loss_pos = (
            y
            * torch.pow(1.0 - xs_pos, self.gamma_pos)
            * torch.log(xs_pos.clamp(min=self.eps))
        )

        # --- Negative Component ---
        # ASL modification: Shifted probability for negatives
        # p_m = max(p - clip, 0)
        # This hard-thresholds easy negatives (where p < clip) to have 0 loss and 0
    gradient
        xs_neg = xs_pos
        if self.clip > 0:
            xs_neg = (xs_neg - self.clip).clamp(min=0)

        # Negative term: -(1-y) * (p_m)^gamma_neg * log(1 - p_m)
        loss_neg = (
            (1.0 - y)
            * torch.pow(xs_neg, self.gamma_neg)
            * torch.log((1.0 - xs_neg).clamp(min=self.eps))
        )

        # Combine components
        # Note: The negative signs from the formulas are applied here
        loss = -(loss_pos + loss_neg)

        # Apply reduction
        if self.reduction == "mean":
            return loss.mean()
        elif self.reduction == "sum":
            return loss.sum()
        else:
            return loss
```

## Module: library/models.py

```python
import timm
import torch.nn as nn
from library.config import Config

def get_model(model_name, num_classes=Config.NUM_CLASSES, pretrained=True):
    """
    Creates and returns a model architecture based on the provided name using the
    timm library.

    This function instantiates the backbone (e.g., ResNet101d, ConvNeXt-Base) and
    replaces the classification head to match the specific number of classes in the
    dataset.
    It adheres to the strategy of using Global Average Pooling followed by a Linear
    projection,
    which is the default behavior of timm's create_model when num_classes is
    specified.

    Args:
        model_name (str): Name of the model architecture (e.g., 'resnet101d', '
    convnext_base').
```

```
        num_classes (int): Number of output classes (attributes). Defaults to Config.
    NUM_CLASSES.
        pretrained (bool): Whether to load pretrained ImageNet weights. Defaults to
    True.

    Returns:
        nn.Module: The instantiated PyTorch model.
    """
    try:
        # Instantiate the model using timm
        # num_classes argument automatically resets the classifier head to the
    correct size
        # and initializes it randomly.
        model = timm.create_model(
            model_name, pretrained=pretrained, num_classes=num_classes
        )

        return model

    except Exception as e:
        raise RuntimeError(f"Failed to create model '{model_name}' using timm: {e}")
```

## Module: library/train.py

```python
import os
import time
import torch
import torch.nn as nn
import torch.optim as optim
import numpy as np
from torch.cuda.amp import GradScaler, autocast
from torch.optim.lr_scheduler import OneCycleLR

from library.config import Config, seed_everything
from library.dataset import get_dataloaders
from library.models import get_model
from library.loss import AsymmetricLoss
from library.utils import calculate_f1_score, optimize_threshold, save_checkpoint

class Trainer:
    """
    Manages the training and validation process for a single model.
    """

    def __init__(self, model, optimizer, scheduler, criterion, device, scaler):
        self.model = model
        self.optimizer = optimizer
        self.scheduler = scheduler
        self.criterion = criterion
        self.device = device
        self.scaler = scaler
        self.best_score = 0.0

    def train_epoch(self, train_loader, epoch):
        """
        Runs one epoch of training.
        """
        self.model.train()
        running_loss = 0.0
        num_batches = len(train_loader)
```

```python
        for i, (images, targets, _) in enumerate(train_loader):
            images = images.to(self.device, non_blocking=True)
            targets = targets.to(self.device, non_blocking=True)

            self.optimizer.zero_grad()

            # Mixed precision training
            with autocast():
                outputs = self.model(images)
                loss = self.criterion(outputs, targets)

            self.scaler.scale(loss).backward()
            self.scaler.step(self.optimizer)
            self.scaler.update()

            if self.scheduler is not None:
                self.scheduler.step()

            running_loss += loss.item()

        avg_loss = running_loss / num_batches
        return avg_loss

    def validate(self, val_loader):
        """
        Runs validation and calculates metrics.
        """
        self.model.eval()
        running_loss = 0.0
        all_preds = []
        all_targets = []

        with torch.no_grad():
            for images, targets, _ in val_loader:
                images = images.to(self.device, non_blocking=True)
                targets = targets.to(self.device, non_blocking=True)

                # Mixed precision inference
                with autocast():
                    outputs = self.model(images)
                    loss = self.criterion(outputs, targets)

                running_loss += loss.item()

                # Apply sigmoid for probabilities
                probs = torch.sigmoid(outputs)

                all_preds.append(probs.cpu().numpy())
                all_targets.append(targets.cpu().numpy())

        avg_loss = running_loss / len(val_loader)
        all_preds = np.concatenate(all_preds, axis=0)
        all_targets = np.concatenate(all_targets, axis=0)

        # Calculate F1 Score (Standard 0.5 threshold)
        f1_score = calculate_f1_score(all_targets, all_preds, threshold=0.5)

        # Calculate Optimized F1 Score (for monitoring potential)
        best_thresh, opt_f1 = optimize_threshold(all_targets, all_preds, num_steps
=50)

        return avg_loss, f1_score, opt_f1, best_thresh
```

```python
    def fit(self, train_loader, val_loader, epochs, model_name, patience=5):
        """
        Main training loop with early stopping.
        """
        print(f"Starting training for {model_name}...")
        patience_counter = 0

        for epoch in range(1, epochs + 1):
            start_time = time.time()

            # Train
            train_loss = self.train_epoch(train_loader, epoch)

            # Validate
            val_loss, val_f1, opt_f1, best_thresh = self.validate(val_loader)

            elapsed = time.time() - start_time

            # Print full precision metrics
            print(f"Epoch {epoch}/{epochs} | Time: {elapsed:.2f}s")
            print(f"  Train Loss: {train_loss}")
            print(f"  Val Loss:   {val_loss}")
            print(f"  Val F1:     {val_f1}")
            print(f"  Opt F1:     {opt_f1} (Thresh: {best_thresh})")

            # Checkpointing
            # We use the standard F1 (0.5) or Optimized F1?
            # Usually for multi-label with unknown distribution shifts, optimizing
    threshold is safer.
            # However, for consistency with the prompt's metric definition, we track
     improvement on Opt F1
            # as we will calibrate threshold in inference.
            current_score = opt_f1

            if current_score > self.best_score:
                print(
                    f"  Score improved from {self.best_score} to {current_score}.
    Saving checkpoint."
                )
                self.best_score = current_score
                save_checkpoint(
                    self.model,
                    self.optimizer,
                    epoch,
                    self.best_score,
                    f"{model_name}_best.pth",
                )
                patience_counter = 0
            else:
                patience_counter += 1
                print(f"  No improvement. Patience: {patience_counter}/{patience}")

            if patience_counter >= patience:
                print(f"Early stopping triggered at epoch {epoch}.")
                break

        return self.best_score

def train_specific_model(model_name, epochs=Config.EPOCHS, debug=Config.DEBUG):
    """
    Sets up the environment and trains a specific model architecture.
```

```
Args:
    model_name (str): Name of the model to train (e.g., 'resnet101d').
    epochs (int): Number of epochs to train.
    debug (bool): Whether to run in debug mode (subset of data).
"""
# 1. Setup
seed_everything(Config.SEED)
device = torch.device(Config.DEVICE)

print(f"Initializing {model_name} on {device}...")

# 2. Data
# load_cached_data=True ensures we use the parquet cache if available
train_loader, val_loader, _ = get_dataloaders(
    debug=debug,
    batch_size=Config.BATCH_SIZE,
    num_workers=Config.NUM_WORKERS,
    load_cached_data=True,
)

# 3. Model
model = get_model(model_name, num_classes=Config.NUM_CLASSES, pretrained=True)
model = model.to(device)

# 4. Optimizer & Scheduler
optimizer = optim.AdamW(
    model.parameters(), lr=Config.LEARNING_RATE, weight_decay=Config.
WEIGHT_DECAY
)

# OneCycleLR needs total steps
steps_per_epoch = len(train_loader)
scheduler = OneCycleLR(
    optimizer,
    max_lr=Config.LEARNING_RATE,
    epochs=epochs,
    steps_per_epoch=steps_per_epoch,
    pct_start=0.1,  # Warmup for first 10%
    div_factor=25.0,
    final_div_factor=1000.0,
)

# 5. Loss & Scaler
criterion = AsymmetricLoss(
    gamma_neg=Config.ASL_GAMMA_NEG,
    gamma_pos=Config.ASL_GAMMA_POS,
    clip=Config.ASL_CLIP,
)
scaler = GradScaler()

# 6. Trainer
trainer = Trainer(model, optimizer, scheduler, criterion, device, scaler)

# 7. Execute
best_score = trainer.fit(train_loader, val_loader, epochs, model_name)

print(f"Training finished for {model_name}. Best F1 Score: {best_score}")

# Clear memory
del model, optimizer, scheduler, scaler, trainer, train_loader, val_loader
torch.cuda.empty_cache()
```

```
        return best_score
```

## Module: library/utils.py

```python
import os
import torch
import numpy as np
from sklearn.metrics import f1_score
from library.config import Config

def calculate_f1_score(y_true, y_pred, threshold=0.5):
    """
    Calculates the Micro-averaged F1 score.

    Args:
        y_true (np.ndarray or torch.Tensor): Ground truth labels (N, C).
        y_pred (np.ndarray or torch.Tensor): Predicted probabilities (N, C).
        threshold (float): Threshold for binarizing predictions.

    Returns:
        float: Micro F1 score.
    """
    # Ensure inputs are numpy arrays
    if isinstance(y_true, torch.Tensor):
        y_true = y_true.detach().cpu().numpy()
    if isinstance(y_pred, torch.Tensor):
        y_pred = y_pred.detach().cpu().numpy()

    # Binarize predictions based on the threshold
    y_pred_bin = (y_pred > threshold).astype(int)

    # Calculate Micro-F1 score
    return f1_score(y_true, y_pred_bin, average="micro")

def optimize_threshold(y_true, y_pred, num_steps=100):
    """
    Finds the optimal decision threshold for Micro F1 score via linear search.

    Args:
        y_true (np.ndarray or torch.Tensor): Ground truth labels.
        y_pred (np.ndarray or torch.Tensor): Predicted probabilities.
        num_steps (int): Number of steps in the linear search (between 0 and 1).

    Returns:
        tuple: (best_threshold, best_score)
    """
    if isinstance(y_true, torch.Tensor):
        y_true = y_true.detach().cpu().numpy()
    if isinstance(y_pred, torch.Tensor):
        y_pred = y_pred.detach().cpu().numpy()

    best_threshold = 0.5
    best_score = -1.0

    # Search range from 0.01 to 0.99 to avoid edge cases
    thresholds = np.linspace(0.01, 0.99, num_steps)

    for thresh in thresholds:
        score = calculate_f1_score(y_true, y_pred, threshold=thresh)
```

```
        if score > best_score:
            best_score = score
            best_threshold = thresh

    return best_threshold, best_score

def save_checkpoint(model, optimizer, epoch, score, filename):
    """
    Saves a model checkpoint to the working directory.

    Args:
        model (torch.nn.Module): The model to save.
        optimizer (torch.optim.Optimizer): The optimizer state.
        epoch (int): Current epoch.
        score (float): Validation score (F1).
        filename (str): Name of the file to save.
    """
    # Ensure the working directory exists
    os.makedirs(Config.WORKING_DIR, exist_ok=True)

    filepath = os.path.join(Config.WORKING_DIR, filename)

    state = {
        "epoch": epoch,
        "model_state_dict": model.state_dict(),
        "optimizer_state_dict": (
            optimizer.state_dict() if optimizer is not None else None
        ),
        "score": score,
    }

    torch.save(state, filepath)

def load_checkpoint(model, filename, optimizer=None, device=Config.DEVICE):
    """
    Loads a model checkpoint.

    Args:
        model (torch.nn.Module): The model to load weights into.
        filename (str): Name of the checkpoint file (relative to WORKING_DIR or
    absolute).
        optimizer (torch.optim.Optimizer, optional): Optimizer to load state into.
        device (str): Device to map location to.

    Returns:
        tuple: (epoch, score)
    """
    # Determine full path
    if os.path.exists(filename):
        filepath = filename
    else:
        filepath = os.path.join(Config.WORKING_DIR, filename)

    if not os.path.exists(filepath):
        raise FileNotFoundError(f"Checkpoint file not found: {filepath}")

    # Load checkpoint
    checkpoint = torch.load(filepath, map_location=device)

    # Load model state
    model.load_state_dict(checkpoint["model_state_dict"])
```

```
        # Load optimizer state if provided and available
        if optimizer is not None and checkpoint.get("optimizer_state_dict") is not None:
            optimizer.load_state_dict(checkpoint["optimizer_state_dict"])

        epoch = checkpoint.get("epoch", 0)
        score = checkpoint.get("score", 0.0)

        return epoch, score
```

## Main Script: main.py

```python
import sys
import os
import numpy as np
import pandas as pd
import torch
import cv2
from library.config import Config, seed_everything
from library.train import train_specific_model
from library.inference import (
    get_model_predictions,
    ensemble_predictions,
    generate_submission,
)
from library.utils import optimize_threshold
from library.dataset import get_dataloaders

def main():
    # 1. Setup
    seed_everything(Config.SEED)
    device = torch.device(Config.DEVICE)

    print("=== Starting Orchestration Script ===")

    # 2. Train Models
    # We train for 5 epochs to improve convergence while staying within the 4-hour
    limit.
    # Cite Lesson 00003: Longer schedule allows better convergence.
    # Cite Lesson 00018: Ensemble of diverse models.
    print(f"\n[Training] Starting training for Model A: {Config.MODEL_A_NAME}")
    train_specific_model(Config.MODEL_A_NAME, epochs=5, debug=False)

    print(f"\n[Training] Starting training for Model B: {Config.MODEL_B_NAME}")
    train_specific_model(Config.MODEL_B_NAME, epochs=5, debug=False)

    # 3. Validation & Metric Calculation
    print("\n[Validation] Loading validation data...")
    # We only need the validation loader here to get targets and run inference
    _, val_loader, _ = get_dataloaders(debug=False, load_cached_data=True)

    print("[Validation] Generating ensemble predictions...")
    # Get predictions for Model A
    # load_cached_data=False ensures we use the newly trained weights
    probs_a, targets = get_model_predictions(
        Config.MODEL_A_NAME, "val", val_loader, device, load_cached_data=False
    )
    # Get predictions for Model B
    probs_b, _ = get_model_predictions(
        Config.MODEL_B_NAME, "val", val_loader, device, load_cached_data=False
```

```
)

# Ensemble predictions (simple average)
ensemble_probs = ensemble_predictions([probs_a, probs_b])

# Ensure targets are numpy array
if isinstance(targets, torch.Tensor):
    targets = targets.cpu().numpy()
elif isinstance(targets, list):
    targets = np.array(targets)

# Optimize threshold
best_thresh, best_score = optimize_threshold(targets, ensemble_probs)

# REQUIRED OUTPUT
print(f"Final Validation Metric: {best_score}")

# 4. Failure Analysis
print("\n[Analysis] Performing Failure Analysis...")

# Calculate per-sample F1 to determine error magnitude
# Binarize predictions using the optimal threshold
preds_bin = (ensemble_probs > best_thresh).astype(int)

# Calculate F1 per sample (instance-level)
# F1 = 2*TP / (2*TP + FP + FN)
tp = np.sum((preds_bin == 1) & (targets == 1), axis=1)
fp = np.sum((preds_bin == 1) & (targets == 0), axis=1)
fn = np.sum((preds_bin == 0) & (targets == 1), axis=1)

epsilon = 1e-7
f1_samples = (2 * tp) / (2 * tp + fp + fn + epsilon)
error_magnitude = 1.0 - f1_samples

# Load metadata to get features
val_df = pd.read_csv(Config.VAL_CSV)

# Feature 1: Label Cardinality (from ground truth)
# Handle potential NaNs in attribute_ids
val_df["attribute_ids"] = val_df["attribute_ids"].fillna("")
val_df["num_labels"] = val_df["attribute_ids"].apply(
    lambda x: len(x.split()) if x.strip() else 0
)

# Feature 2: Image Brightness (from image content)
# We process a subset of validation images to save time
sample_size = min(2000, len(val_df))
sample_indices = np.random.choice(len(val_df), size=sample_size, replace=False)

brightness_values = []
sampled_errors = []
sampled_cardinality = []

print(f"  Processing {sample_size} images for feature extraction...")
for idx in sample_indices:
    row = val_df.iloc[idx]
    path = os.path.join(Config.INPUT_DIR, row["file_path"])

    # Read image
    img = cv2.imread(path)
    if img is not None:
        # Calculate mean brightness
        b = np.mean(img) / 255.0
```

```
                brightness_values.append(b)
                sampled_errors.append(error_magnitude[idx])
                sampled_cardinality.append(val_df.iloc[idx]["num_labels"])

        # Calculate Correlations
        if len(sampled_errors) > 1:
            # Correlation with Label Cardinality
            corr_card = np.corrcoef(sampled_cardinality, sampled_errors)[0, 1]
            print(f"Correlation (Error vs Label Cardinality): {corr_card:.4f}")

            # Correlation with Brightness
            corr_bright = np.corrcoef(brightness_values, sampled_errors)[0, 1]
            print(f"Correlation (Error vs Image Brightness): {corr_bright:.4f}")
        else:
            print("Insufficient data for correlation analysis.")

        # 5. Submission
        THRESHOLD_SCORE = 0.6335639211171432

        print(f"\n[Submission] Checking threshold: {best_score} > {THRESHOLD_SCORE}")

        if best_score > THRESHOLD_SCORE:
            print("Threshold met. Generating submission file...")
            # generate_submission handles test inference and saving
            # load_cached_data=False ensures we generate fresh test predictions
            generate_submission(debug=False, load_cached_data=False)
        else:
            print("Threshold not met. Skipping submission generation.")

        print("\n=== Workflow Completed ===")

if __name__ == "__main__":
    main()
```

