# OpenReview forum: "MARS: Modular Agent with Reflective Search for Automated AI Research"
_ICML.cc/2026/Conference — ICML 2026 regular_

### Official Review · Reviewer_ZXjX · 2026-03-08

**Soundness:** 3
**Presentation:** 3
**Significance:** 2
**Originality:** 2
**Overall Recommendation:** 4
**Confidence:** 3

**Summary:**

This paper proposes MARS, an LLM-based agent framework for automated machine learning engineering. It has three main components: (1) budget-aware MCTS that penalizes costly solutions during tree search, (2) a modular code generation pipeline that produces multi-file repositories instead of monolithic scripts, and (3) a comparative reflective memory that distills lessons from pairwise comparisons between solutions. The system is evaluated on MLE-Bench and achieves strong medal rates compared to existing open-source agents like AIDE and AIRA-dojo.

**Compliance With Llm Reviewing Policy:**

Affirmed.

**Final Justification:**

I recommend 4: Weak accept. The paper presents a solid and practically meaningful system for autonomous ML engineering, with careful evaluation and strong benchmark results. My main concerns were whether the gains were mainly due to higher budget and whether the ablations on Lite alone were sufficient. The rebuttal addressed both points reasonably well: the added cost-controlled comparison suggests the improvements are not simply from spending more, and the new Medium/High subset ablations make the component analysis more convincing.

I still think the methodological novelty is moderate, since the main ingredients are individually fairly standard, and the added ablations are not yet full-benchmark. However, the rebuttal improved my confidence in the paper’s soundness and significance, so overall I am updating my recommendation from weak reject to weak accept.

**Key Questions For Authors:**

1. Have you tried running AIDE or AIRA-dojo with comparable API budget or with the 2×A100 setup used for MARS+? The current comparison controls hardware but not API cost. If baselines close the gap when given the same monetary budget, it would undermine the claim that MARS's architectural choices are responsible for the gains.

2. Can you run the ablation study on the full 75-task benchmark, or at least report ablation results broken down by Lite/Medium/High? If modular decomposition and lesson learning show diminishing returns on harder tasks, that would change how I evaluate the contribution. Conversely, if gains are consistent across difficulty levels, that would strengthen the paper.

3. Do you have any analysis (even manual inspection on a subset) of whether the distilled lessons are actually causally correct? This directly relates to whether the system is truly solving credit assignment or just generating plausible-sounding narratives. If the authors can show concrete evidence of causal accuracy, I would consider raising my score.

**Limitations:**

yes

**Strengths And Weaknesses:**

### Strengths
1. The problem framing is reasonable and practical. The observation that MLE differs from standard SWE tasks due to expensive evaluations, and the three-pillar decomposition follows naturally from these observations.
2. The controlled evaluation is done carefully. Running AIDE and AIRA-dojo under identical hardware and with the same LLMs is exactly the right thing to do for this kind of benchmark.
3. The qualitative analysis is interesting. The lesson-transfer rate is a nice metric that provides evidence the reflective memory is doing something meaningful beyond just caching within a single trajectory.
4. Ablations cover the main components. Figures 3 and 4 show that removing modular decomposition, lesson learning, or swapping MCTS variants each degrades performance.


### Weaknesses
1.  The three components are individually quite standard. The budget-aware reward is a weighted product borrowed from MnasNet; modular code generation is prompt engineering with role-specialized agents; comparative reflective memory amounts to asking the LLM to diff two solutions and write down what changed. The system is well-engineered, but I struggle to identify a component that offers methodological insight beyond sensible system design.

2. Ablations are only on MLE-Bench Lite, which is the easiest split. Conclusions from easy tasks may not transfer, for example, does modular decomposition even matter on Lite where monolithic scripts are probably fine? Ablations across all three difficulty levels would be much more convincing.

3. MARS costs 60.5, while 39.0 for AIRA-dojo and10.6 for AIDE. This raises a natural question: would simply giving the baselines comparable API budget (more iterations, or the 2×A100 setup used for MARS+) close the gap? Without this control, it's hard to separate the effect of architecture from the effect of spending more.

---

> ### Author Rebuttal · Authors · 2026-03-31
>
> ## W1: Methodology Novelty
>
> We appreciate the reviewer’s observation. While MARS builds on established concepts, its methodological insight lies in the principled synergy of its closed-loop architecture, specifically engineered to resolve the credit assignment problem in resource-constrained MLE:
>
> - MARS enforces a Modular paradigm to enable precise MCTS navigation at the repository level. This bypasses the structural fragility and context-noise typical of monolithic agents, which we identify as a primary bottleneck in autonomous engineering.
> - Unlike standard memory mechanisms, Comparative Reflective Memory leverages the branching structure of MCTS to isolate specific causal factors. The 63% cross-branch transfer rate is a non-trivial result, proving the framework's emergent ability to generalize "Aha!" moments across independent exploration paths.
>
> The contribution is thus the systematic integration of budget-aware planning and repository-level reflection to enable SOTA performance in the domain of MLE.
>
> ---
>
> ## W2&Q2: Ablations across all difficulty levels of MLE-Bench would be more convincing.
>
> We appreciate the reviewer’s request to validate our framework’s components across varying levels of task complexity. While our initial evaluation focused on the **MLE-Bench Lite** split (22 tasks), we have extended the ablation study to include a random sample of 10 tasks each from the **Medium** and **High** splits. The results (detailed in https://tinyurl.com/5t4msvdw - see Figure 1-12) confirm that the benefits of our core modules are consistent – and often more pronounced – as difficulty increases.
>
> ---
>
> ## W3&Q1: Would simply giving the baselines comparable API budget close the gap?
>
> We thank the reviewer for this critical question. To demonstrate that MARS's gains are driven by architectural efficiency rather than a higher financial budget, we conducted a cost-controlled evaluation. The results in the following table demonstrate that MARS achieves a higher "Any Medal Rate" while spending **less** than both AIDE and AIRA-dojo. This "Pareto improvement" – achieving better results with a lower budget – directly substantiates our claim that MARS's specific architectural choices are responsible for the performance gains, not simply a larger resource allocation.
>
> **Table:** Compare MARS with the baselines under the same hardware settings and comparable API costs. MARS is run under 4 hour limit while the baselines are run under 24 hour limit.
> | Metric             | AIDE           | AIRA-dojo      | MARS           |
> |--------------------|----------------|----------------|----------------|
> | Price ($)          | 10.6 $\pm$ 0.9 | 39.0 $\pm$ 3.7 | 9.6 $\pm$ 2.2  |
> | Any Medal Rate (%) | 23.1 $\pm$ 0.4 | 24.4 $\pm$ 1.2 | 28.4 $\pm$ 1.2 |
>
> ---
>
> ## Q3: Do you have any analysis of whether the distilled lessons are causally correct?
>
> To verify if MARS performs genuine credit assignment, we conducted a systematic evaluation of all distilled lessons:
>
> - We utilized Claude 4.6 Sonnet to evaluate all 3,611 solution lessons generated by GEMINI-3-Pro-Preview for causal accuracy. The audit confirmed that 88.34% of lessons correctly attributed validation metric shifts to specific code changes, rather than generating hallucinated narratives.
>
> - We manually inspected a representative subset of 20 lessons. This inspection yielded a 90% causal accuracy rate for the subset and confirmed that the LLM-based audit is a reliable proxy for expert human judgment.
>
> - Ex 1: For the tabular-playground-series-may-2022 task, MARS diagnosed a 0.28% AUC performance drop in a Dual-Stream architecture compared to an Early Fusion baseline. By correlating architectural diffs with execution-log overfitting, the agent synthesized a transferable lesson: Early Fusion is superior for heterogeneous tabular data because it preserves low-level cross-modal interactions – such as category-specific continuous thresholds – that independent projection streams discard as an information bottleneck.
>
> - Ex 2: For the Histopathologic Cancer Detection task, MARS diagnosed why a multi-model ensemble underperformed (0.976 AUC) against its own single-model component (0.982 AUC). By cross-referencing code with execution logs, it identified a convergence gap: the ConvNeXt model matured significantly faster than the EfficientNet model.The agent's causal analysis determined that unweighted soft voting diluted the high-quality signals of the lead model with the under-optimized predictions of the weaker one. Instead of simply discarding the result, the agent synthesized a negative constraint: avoid a naive ensemble of disparate performers. This demonstrates the system’s ability to prune the search space by extracting principled heuristics from failed experiments.
>
> This analysis provides concrete evidence that Comparative Reflective Memory effectively resolves the credit assignment problem. We will include this detailed evaluation and qualitative examples in the final version.

---

> > ### Author Rebuttal · Reviewer_ZXjX · 2026-04-02
> >
> > I appreciate the added ablation claim beyond the Lite split. However, the detailed results are provided only through an external short URL that appears to redirect to Google Drive, which is not ideal for double-blind reviewing. As such, I only partially update my assessment based on this evidence and would prefer the key quantitative numbers to be summarized directly in the rebuttal.

---

> > > ### Author Response · Authors · 2026-04-02
> > >
> > > We sincerely thank the reviewer for the positive feedback and for acknowledging that their concerns have been fully resolved. We apologize for the oversight regarding the external URL; We understand this is not ideal for the double-blind review process.
> > >
> > > As requested, we have converted the key quantitative results from those figures into the markdown tables below. These results demonstrate the consistent performance gains of our proposed components.
> > >
> > > ---
> > >
> > > **Table 1:** Impact of memory mechanisms for MARS.
> > > | Split | Method | Any Medal Rate at 24 hour (%) |
> > > |-------|--------|---------------------|
> > > | Lite | MARS without Any Memory | 51.52 ± 1.52 |
> > > | Lite | MARS with only Reflective Memory | 62.12 ± 1.52 |
> > > | Lite | MARS with Comparative Reflective Memory (ours) | **74.24** ± 1.52 |
> > > | Medium subset | MARS without Any Memory | 10.00 ± 0.00 |
> > > | Medium subset | MARS with only Reflective Memory | 20.00 ± 5.77 |
> > > | Medium subset | MARS with Comparative Reflective Memory (ours) | **56.67** ± 8.82 |
> > > | High subset | MARS without Any Memory | 20.00 ± 5.77 |
> > > | High subset | MARS with only Reflective Memory | 26.67 ± 3.33 |
> > > | High subset | MARS with Comparative Reflective Memory (ours) | **46.67** ± 3.33 |
> > >
> > > ---
> > >
> > > **Table 2:**  Comparison of tree search strategies for MARS.
> > > | Split | Method | Any Medal Rate at 24 hour (%) |
> > > |-------|--------|---------------------|
> > > | Lite | Greedy Search | 59.09 ± 5.25 |
> > > | Lite | Vanilla MCTS (w=0) | 63.64 ± 2.62 |
> > > | Lite | Budget-aware MCTS (ours) | **74.24** ± 1.52 |
> > > | Medium subset | Greedy Search | 13.33 ± 8.82 |
> > > | Medium subset | Vanilla MCTS (w=0) | 30.00 ± 5.77 |
> > > | Medium subset | Budget-aware MCTS (ours) | **56.67** ± 8.82 |
> > > | High subset | Greedy Search | 36.67 ± 8.82 |
> > > | High subset | Vanilla MCTS (w=0) | 30.00 ± 5.77 |
> > > | High subset | Budget-aware MCTS (ours) | **46.67** ± 3.33 |
> > >
> > > ---
> > >
> > > **Table 3:** Impact of the penalty weight $w$ on the performance of MARS.
> > >
> > > | Split | Method | Any Medal Rate at 24 hour (%) |
> > > |-------|--------|---------------------|
> > > | Lite | Budget-agnostic MCTS (w=0) | 63.64 ± 2.62 |
> > > | Lite | Budget-aware MCTS (w=-0.15) | 53.03 ± 1.52 |
> > > | Lite | Budget-aware MCTS (w=-0.07, ours) | **74.24** ± 1.52 |
> > > | Medium subset | Budget-agnostic MCTS (w=0) | 30.00 ± 5.77 |
> > > | Medium subset | Budget-aware MCTS (w=-0.15) | 23.33 ± 3.33 |
> > > | Medium subset | Budget-aware MCTS (w=-0.07, ours) | **56.67** ± 8.82 |
> > > | High subset | Budget-agnostic MCTS (w=0) | 30.00 ± 5.77 |
> > > | High subset | Budget-aware MCTS (w=-0.15) | 23.33 ± 3.33 |
> > > | High subset | Budget-aware MCTS (w=-0.07, ours) | **46.67** ± 3.33 |
> > >
> > > ---
> > >
> > > **Table 4:** Impact of Modular Decomposition for MARS.
> > >
> > > | Split | Method | Any Medal Rate at 24 hour (%) |
> > > |-------|--------|---------------------|
> > > | Lite | MARS without Modular Decomposition | 62.12 ± 4.01 |
> > > | Lite | MARS (ours) | **74.24** ± 1.52 |
> > > | Medium subset | MARS without Modular Decomposition | 33.33 ± 3.33 |
> > > | Medium subset | MARS (ours) | **56.67** ± 8.82 |
> > > | High subset | MARS without Modular Decomposition | 33.33 ± 8.82 |
> > > | High subset | MARS (ours) | **46.67** ± 3.33 |

---

### Official Review · Reviewer_wqLv · 2026-03-08

**Soundness:** 3
**Presentation:** 3
**Significance:** 3
**Originality:** 3
**Overall Recommendation:** 5
**Confidence:** 4

**Summary:**

The paper introduces MARS (Modular Agent with Reflective Search), an LLM-based agentic framework designed for automated Machine Learning Engineering (MLE) and AI research. The authors identify three critical bottlenecks in applying existing coding agents to MLE: the generation of fragile monolithic scripts, the high computational cost of evaluating ML pipelines, and the difficulty of credit assignment when performance changes. To tackle these, MARS proposes a "Design-Decompose-Implement" pipeline to enforce modular codebase generation, a Budget-Aware MCTS that incorporates execution latency into the search reward, and a Comparative Reflective Memory to extract structured lessons from both successful and failed runs. The framework is evaluated on the MLE-Bench benchmark, where it achieves state-of-the-art results among open-source frameworks in controlled settings and demonstrates competitive performance against top leaderboard methods.

**Compliance With Llm Reviewing Policy:**

Affirmed.

**Final Justification:**

FInal Justification: Accept.

The authors have solved my concerns during rebuttal phase.

**Key Questions For Authors:**

Generalization to other SOTA Models: The impressive results rely on Gemini-2.5/3 Pro models. Have you experimented with other state-of-the-art models to see if they possess the necessary reasoning capabilities to handle the strict modular decomposition and lesson distillation prompts?

**Limitations:**

1. Benchmark  Generalization: Currently, the framework's effectiveness is only demonstrated on MLE-Bench, which predominantly consists of Kaggle-style predictive modeling tasks. To further validate the paper's purpose of "Automating AI Research," the authors might evaluate MARS on other benchmarks that test different stages of the research lifecycle. For example, testing the agent on PaperBench would provide crucial insights into whether the framework can genuinely replicate and implement broader AI research papers beyond structured competitions.

**Strengths And Weaknesses:**

Strengths:
1. Budget aware MCTS is really interesting. This MARS framework balances the cost and exploration. Meanwhile, keep learned lessons in good arrangement.
2. Cost-Aware Search Strategy: Integrating execution time into the MCTS objective (Budget-Aware MCTS) is a clever solution for MLE tasks. Standard search algorithms often fail in ML domains because evaluating a node can take hours; penalizing high-latency solutions forces the agent to explore more efficient baselines first.
3. Solid Empirical Evaluation and Ablation: The paper presents rigorous experiments on a challenging benchmark. The ablation studies effectively isolate the contributions of the modular design and the lesson learning mechanisms, proving their necessity.

Weaknesses:
1. Slightly Overstated Framing: Framing the paper around "Automated AI Research" feels slightly overstated. The MLE-Bench tasks are essentially Kaggle competitions focusing on data preprocessing, hyperparameter tuning, and ensembling. While impressive, this is distinct from open-ended, foundational AI research (e.g., proposing novel architectures or loss functions).

---

> ### Author Rebuttal · Authors · 2026-03-31
>
> ## W1: Slightly Overstated Framing
>
> We thank the reviewer for the feedback on our framing. We will update the paper to focus specifically on **Autonomous MLE**, as our evaluation on MLE-Bench primarily substantiates effectiveness in the engineering-heavy experimental phase of the research cycle. While we view MLE as a critical "experimental backbone" for AI research, we will clarify this scope in the revised manuscript and discuss the transition to open-ended AI research as a distinct future direction.
>
> ---
>
> ## Q1: Generalization to other SOTA Models
>
> We appreciate the reviewer’s insightful question regarding the framework's dependency on specific LLM providers. To demonstrate that the success of MARS is a result of its **architectural robustness** rather than model-specific quirks, we conducted a cross-model evaluation using **Claude 4.6 Sonnet**. The results in the following table confirm that MARS maintains a dominant performance margin even when powered by a different reasoning engine.
>
> **Table:** Model generalizability on MLE-Bench using Claude 4.6 Sonnet. Values represent Any Medal Rate (Mean $\pm$ SEM) over three independent runs, evaluated on a random sample ($n=10$ per split) of Lite, Medium, and High tasks.
> | Method      | Lite          | Medium       | High         | All          |
> |-------------|--------------|--------------|--------------|--------------|
> | AIRA-dojo   | 40.0$\pm$5.8 | 6.7$\pm$3.3  | 13.3$\pm$3.3 | 20.0$\pm$1.9 |
> | MARS (ours) | 80.0$\pm$0.0 | 43.3$\pm$3.3 | 43.3$\pm$3.3 | 55.6$\pm$1.1 |
>
> ---
>
> ## L1: Benchmark Generalization
>
> We appreciate the reviewer’s feedback regarding the scope of our evaluation. We acknowledge that "Automated AI Research" encompasses a wide spectrum of tasks beyond predictive modeling. To address this, we will:
>
> - **Refine Framing:** Update the paper to focus specifically on the engineering-heavy experimental phase of the research cycle (MLE).
> - **Discuss Generalizability:** In the Conclusion section, we will explicitly discuss how the "Design-Decompose-Implement" paradigm and Comparative Reflective Memory could be adapted for broader research benchmarks like PaperBench. We will frame the transition from structured competition environments to open-ended, multi-stage research replication as a primary objective for future development.
>
> This adjustment ensures our claims are strictly substantiated by the MLE-Bench results while providing a clear roadmap for generalizing MARS to the broader scientific discovery lifecycle.

---

> > ### Author Rebuttal · Reviewer_wqLv · 2026-04-03
> >
> > Thank you for your detailed response. I think the main concerns have been solved in your rebuttal. I will maintain my score because this score has been very positive.

---

### Official Review · Reviewer_DUZg · 2026-03-13

**Soundness:** 3
**Presentation:** 3
**Significance:** 3
**Originality:** 3
**Overall Recommendation:** 4
**Confidence:** 4

**Summary:**

This work proposes a "modular agent with reflective search" framework for autonomous AI research. The key idea is (1) Budget-Aware MCTS, (2) Modular Construction with "Design-Decompose-Implement", and (3) Lesson learning from previous memory.
The authors validate the proposed approach in MLE-Bench, substantially improving previous baselines such as AIDE and AIRA-dojo.

**Compliance With Llm Reviewing Policy:**

Affirmed.

**Final Justification:**

My concerns have been adequately addressed.

**Key Questions For Authors:**

1. Can authors elaborate more on why "Modular Construction" is useful? Since such a modular idea does not seem to be a bottleneck for current coding agents.

**Limitations:**

The authors might want to test the framework in more open-ended real research tasks.

**Strengths And Weaknesses:**

Automating AI research, or technically, machine learning engineering, is a very important problem to study. This work, in general, offers valuable insights along this direction. I personally find the Budget-Aware MCTS pretty interesting, as how to balance resources is practically very important for GPU-intensive research.

There are a few limitations.

1. The framing can be more accurate. I won't call solving MLE Bench AI research, which is literally called machine learning engineering. AI research should be more open-ended [1] [2].
2. Only the experiment is done on MLE-Bench, which might raise concerns about how generalizable the conclusion is. For example, rules like "a strict 24-hour wall-clock time budget per competition" might favor algorithms that are aware of the time budget.
3. The naming consistency can be improved, in the abstract, the authors use "Comparative Reflective Memory", in Section 4.3, the section name is "Lesson Learning."

[1] Can LLMs Generate Novel Research Ideas? A Large-Scale Human Study with 100+ NLP Researchers, 2025

[2] Towards Execution-Grounded Automated AI Research, 2026

---

> ### Author Rebuttal · Authors · 2026-03-31
>
> ## W1: Better framing.
>
> We appreciate the reviewer’s feedback regarding the distinction between MLE and open-ended AI research. We re-frame to focus specifically on Autonomous MLE. While we initially positioned MLE as a representative instantiation of the broader AI research process, we agree that our current evaluation on MLE-Bench substantiates effectiveness specifically within the engineering domain. The revised manuscript will clarify this scope and discuss the transition to open-ended AI research as a separate future direction.
>
> ---
>
> ## W2: Rules like "a strict 24-hour wall-clock time budget per competition" might favor algorithms that are aware of the time budget.
>
> We thank the reviewer for the opportunity to clarify the nature of our budget-aware mechanism. We contend that MARS's success is driven by intrinsic efficiency optimization rather than "deadline exploitation".
> - **Proactive Trajectory Improvement:** As shown in **Figure 7**, the performance gap between the budget-aware ($w = -0.07$) and budget-agnostic ($w = 0$) versions emerges as early as **Hour 2**. This confirms the agent proactively prunes low-ROI, high-cost experiments long before the 24-hour limit becomes a factor.
> - **Mathematical Generalizability:** The time limit $L$ in **Eq. 4** serves as a **normalization constant**. Since $L$ is constant for all nodes in a tree, the node ranking depends on the performance-per-unit-compute. This ensures the strategy remains effective across any arbitrary budget (e.g., 2 hours vs. 200 hours) by prioritizing high-efficiency discovery.
> In summary, MARS optimizes the discovery rate throughout the entire search period, making it a generalizable framework for resource-constrained MLE rather than a baseline-specific exploit.
>
> ---
>
> ## W3: Naming consistency
>
> We will change the name of Section 4.3 to "Comparative Reflective Memory".
>
> ---
>
> ## Q1: Why is "Modular Construction" useful? Since such a modular idea does not seem to be a bottleneck for current coding agents.
>
> We acknowledge that while LLMs are capable of modularity, current agentic frameworks such as AIDE and AIRA-dojo for MLE predominantly rely on a **monolithic paradigm**, producing fragile, single-file scripts that are ill-suited for complex MLE repositories. Our Modular Construction pipeline addresses several critical bottlenecks inherent in autonomous MLE (refer to Section 4.2 for details). This structured architecture is what allows MARS to effectively resolve the credit assignment problem by isolating the specific causal factors driving performance shifts.

---

> > ### Author Rebuttal · Reviewer_DUZg · 2026-04-03
> >
> > Thank you for the response! I will keep my positive rating.

---

### Official Review · Reviewer_JnZ4 · 2026-03-13

**Soundness:** 3
**Presentation:** 3
**Significance:** 3
**Originality:** 2
**Overall Recommendation:** 5
**Confidence:** 4

**Summary:**

This paper introduces MARS, an agent framework designed for automated AI research—specifically Machine Learning Engineering (MLE). The authors model the research process as a search problem within a code repository space and propose three core components: (1) Budget-Aware MCTS—which extends standard Monte Carlo Tree Search by introducing an efficiency-guided reward function, R(v) = G(v)·[t(v)/L(v)]^w, to explicitly balance task performance against execution time; (2) Modular Construction—which employs a three-stage "Design-Decompose-Implement" pipeline to generate structured, multi-file code repositories (averaging 6.7 files and 1,104 lines), thereby replacing traditional single-file script generation and enabling diff-based incremental modifications; and (3) Comparative Reflective Memory—which distills structured "lessons" (including causal analyses) by performing code-level comparisons between the current solution and the optimal solution, storing these lessons in a shared pool for retrieval and reuse in subsequent iterations. Evaluated on MLE-Bench (a benchmark comprising 75 Kaggle competitions), MARS achieves State-of-the-Art (SOTA) performance among open-source frameworks—securing a medal in 43.1% of tasks and a Gold medal in 31.1%—while demonstrating a cross-branch lesson transfer rate of 63%.

**Compliance With Llm Reviewing Policy:**

Affirmed.

**Final Justification:**

Thank you for the detailed rebuttal and the additional experiments. The new ablation comparing Comparative Reflective Memory against the Empirical Analysis fallback was very helpful. The distinctions from ExpeL are also now clearer, though I encourage the authors to make these explicit in the revised related work section rather than leaving them only in the rebuttal.  Overall, the rebuttal has addressed the major gaps I identified, and I am happy to raise my score

**Key Questions For Authors:**

1. **Could you provide an ablation study comparing "comparative distillation" against "simple execution log storage"?** This is a critical experiment for validating the core claims of the *Comparative Reflective Memory* component. Currently, the results only demonstrate that "having lessons is better than having none," but they do not prove that "comparative distillation is superior to simple log recording." The system's Appendix F already describes an "Empirical Analysis" mode (which serves as a fallback when no "best solution" is available); this mode could be directly utilized as a baseline for this comparison.

2. **What are the key differences and advantages compared to ExpeL (AAAI 2024)?** ExpeL also distills semantic insights from comparisons between successful and failed attempts and maintains an "insight pool." Please explicitly articulate MARS's specific incremental contributions regarding the granularity of comparison (code diffs vs. action trajectories), the degree of lesson structuring, and cross-branch transfer capabilities—ideally supported by experimental comparisons.

3. **How ​​is the quality of the "lessons" evaluated?** The paper reports a *utilization rate* of 65.8% and a *transfer rate* of 63.0%; however, being "utilized" does not necessarily equate to being "useful." Are there any instances where a lesson was referenced but actually led to a decline in performance? How is the accuracy or reliability of these lessons measured?

4. **How ​​does the approach perform on research tasks outside of the MLE-Bench suite?** The modular code generation paradigm (Design-Decompose-Implement) aligns very naturally with Kaggle-style ML pipelines; however, is it equally applicable to AI research tasks that require mathematical derivation, theoretical analysis, or complex experimental design?

5. **Does the time penalty term *w* in the Budget-Aware MCTS require tuning for different tasks?** The current setting of *w* = −0.07 appears optimal for MLE-Bench as a whole (Figure 7); however, do tasks of varying complexity (Lite / Medium / High) necessitate different values ​​for *w*? If such task-specific tuning is required, this would limit the practical utility of the proposed method.

**Limitations:**

In the Impact Statement, the authors discuss the risks associated with LLMs generating incorrect code, as well as the mitigation strategies involving self-correction via execution feedback. However, the following key limitations have not been sufficiently addressed:

- **The "comparative" mechanism within Reflective Memory was not independently validated**—this constitutes a significant methodological limitation that should be explicitly acknowledged.
- **Evaluation was conducted solely on MLE-Bench**—the scope of evaluation is too narrow to substantiate the broad claims regarding "Automated AI Research."
- **Cost Considerations**—the cost of $60.5 per competition serves as a barrier to widespread adoption; potential avenues for cost reduction (e.g., context caching, early stopping) require discussion.
- **Reliability of "Lessons"**—lessons distilled by LLMs may contain hallucinations or misattributions; the paper fails to address this inherent risk.

It is recommended that a dedicated "Limitations" subsection be added to the camera-ready version of the paper.

**Strengths And Weaknesses:**

**Strengths**

**S1. Clear Problem Definition and Comprehensive System Design.** The paper accurately identifies three key distinctions between automated AI research and general software engineering—namely, high evaluation costs, difficult performance attribution, and high code complexity—and designs three specific components to address them. The overall system design is logically consistent, and the narrative flows smoothly from problem formalization (§3) to methodology (§4) and finally to experimentation (§5).

**S2. Modular Construction is Well-Designed and Empirically Supported.** Elevating LLM-based code generation from single-file scripts to multi-file repositories constitutes a highly valuable engineering contribution. The comparison in Table 4 (1.0 files / 474.8 lines → 6.7 files / 1103.9 lines) and the ablation study in Figure 3 both demonstrate the significant benefits of modularity for complex tasks. The use of diff-based editing avoids the waste associated with full-scale regeneration, representing a practical and reproducible design choice.

**S3. Rigorous Experimental Evaluation and High Reproducibility.** The evaluation is conducted under the standard MLE-Bench protocol, utilizing the same LLMs (Gemini-2.5-Pro/3-Pro-Preview) and open-source baselines (AIDE, AIRA-dojo) to ensure a controlled comparison. Results are reported as the mean ± SEM across three independent runs. Appendix D provides a detailed comparison of hardware configurations across different systems, demonstrating a commitment to ensuring a fair comparison. Appendix F makes all agent prompts publicly available, thereby facilitating reproducibility.

**S4. Budget-Aware MCTS is Simple yet Effective.** Although the time penalty term within the reward function is simple (w = −0.07), the experimental evidence presented in Figures 4 and 5 confirms its effectiveness: the proportion of valid solutions increased from 16.1% (Vanilla MCTS) to 19.5%, and the method consistently outperformed both Greedy Search and Vanilla MCTS under a 24-hour budget. The sensitivity analysis regarding the hyperparameter *w* (Figure 7) is also sufficiently comprehensive.

**S5. Cross-Branch Knowledge Transfer is an Interesting Finding.** ** The 63% "lesson transfer rate" and the "Aha moments" illustrated in Figure 1 demonstrate that the agent is indeed capable of synthesizing knowledge across different search paths; this offers valuable insights into the mechanisms of knowledge accumulation within long-horizon search processes.

**Weaknesses**

**W1. The core claim regarding "Comparative Reflective Memory" lacks experimental validation. [Major]** The paper asserts that the key innovation of its memory mechanism lies in "performing causal attribution via code-level comparisons against the best solution" (which constitutes the meaning of "comparative"). However, the ablation study (Figure 3) merely compares "with lessons vs. without lessons"; it **fails to compare "comparative distillation vs. simple log storage."** The prompts provided in Appendix F reveal that the system possesses a fallback mode (performing "Empirical Analysis" when no "best solution" is available), yet this fallback mode was never utilized as a baseline. Consequently, it remains impossible to determine whether the observed performance improvements stem from the "distillation of lessons themselves" or specifically from the "comparative distillation" process. This constitutes the most significant experimental gap in the paper.

**W2. The discussion of related work concerning memory mechanisms is severely inadequate. [Major]** The "Related Work" section (§2, p.2) devotes a mere four sentences to mentioning Reflexion, Hierarchical Cognitive Caching, and CodeScientist. It completely fails to cite **ExpeL** (AAAI 2024)—a direct conceptual predecessor that similarly employs a process of "comparing successes against failures to distill insights." Furthermore, the paper neglects to discuss other highly relevant works, such as MACLA (AAMAS 2025; comparative process memory refinement), ReasoningBank (Google 2025; strategy-level experience distillation), and A-MEM (NeurIPS 2025; structured associative memory). This oversight renders the paper's positioning regarding novelty imprecise; readers struggle to discern the true incremental contributions of "Comparative Reflective Memory" relative to existing methods for experience distillation.

**W3. The novelty of individual components is limited; the primary contribution lies in the system-level integration. [Moderate]** "Budget-Aware MCTS" essentially amounts to applying a time-based penalty term to the reward function; "Modular Construction" represents the application of standard software engineering practices within an agent-based context; and "Reflective Memory" serves as a specialized adaptation of existing experience distillation paradigms (e.g., ExpeL, Reflexion) tailored to the domain of ML engineering. When viewed in isolation, the novelty of each individual component appears relatively modest. The paper's contribution lies in the effective combination of these three elements; however, presenting a "system combination" as the primary contribution at top-tier conferences typically requires stronger experimental evidence to substantiate the necessity of each individual component. The current ablation study offers only two coarse-grained comparisons (removing modularization / removing lesson learning) and lacks more fine-grained ablations.

**W4. Evaluation is limited to MLE-Bench; generalizability remains questionable. [Moderate]** MLE-Bench consists of ML engineering tasks styled after Kaggle competitions. While the paper's title and abstract claim to target "Automated AI Research," the system has not been evaluated on any other research-oriented tasks (such as RE-Bench, PaperBench, SciCode, etc.). Do modular code generation and lesson learning remain equally effective in non-Kaggle scenarios—such as theoretical derivation, experimental design, or data analysis?

**W5. High operational costs; economic feasibility requires further discussion. [Minor]** Table 10 indicates that the average cost per competition for MARS is $60.5 (compared to $39.0 for AIRA-dojo), driven primarily by the consumption of 286.6 × 10⁵ input tokens (utilized for maintaining the lesson context). Although the paper asserts that "investment yields substantial returns," in practical deployment scenarios, the cumulative cost for 75 competitions amounts to approximately $4,500—a non-negligible expense for academic researchers. The cost-benefit trade-offs and their boundary conditions should be discussed more explicitly.

---

> ### Author Rebuttal · Authors · 2026-03-31
>
> ## W1&Q1: The claim about Comparative Reflective Memory lacks experimental validation.
>
> We thank the reviewer for this insightful suggestion. To isolate the benefit of comparative distillation from general lesson learning, we conducted the ablation across MLE-Bench splits to compare our method against “MARS with only Reflective Memory”, which utilizes the "Empirical Analysis" fallback mode to distill lessons from execution results without comparing them to the best-known solution. The results (shown in https://tinyurl.com/5t4msvdw - Figure 1-3) demonstrate that the "Comparative" component provides a consistent performance boost across all splits. These results confirm that code-level comparisons are essential for isolating causal factors in complex repositories. Without this "delta" analysis, the agent often over-generalizes from noisy logs, whereas “Comparative Reflective Memory” allows it to distill precise, actionable heuristics (mirroring human-led ablation experiments).
>
> ---
>
> ## W2&Q2: The discussion of related work concerning memory mechanisms is inadequate.
>
> We appreciate the reviewer’s suggestion to clarify the positioning of MARS relative to ExpeL and other memory-based agents. We will incorporate citations for ExpeL, MACLA, ReasoningBank, and A-MEM in the revised manuscript.
> The primary distinctions and advantages of MARS over ExpeL include:
> - While ExpeL identifies "good practices" by comparing high-level action trajectories , MARS operates at the repository level, performing code-level diff comparisons. This allows MARS to isolate the specific causal factors driving performance shifts within complex codebases.
> - ExpeL focuses on binary success vs. failure trajectories. In contrast, MARS compares two valid solutions with differing validation metrics. By attributing the metric delta to specific algorithmic changes, MARS can refine solutions that are already functional but sub-optimal.
> - MARS distills structured lessons consisting of algorithmic changes, impact analysis, and generalized rules.
>
> ---
>
> ## W3: The novelty of individual components is limited.
> While we acknowledge that MARS integrates established paradigms, we argue its primary contribution lies in the synergistic framework specifically engineered for the non-binary, resource-intensive nature of MLE. Unlike standard software engineering, MLE requires balancing performance gains against extreme computational costs and opaque credit assignment.
> To substantiate the necessity of each component, we provide fine-grained ablations (See our response to **W1** and our response to Reviewer ZXjX (for W1)). These experiments demonstrate that MARS is not merely a combination of parts, but a co-dependent system where each module provides the necessary grounding for the others to function effectively in the MLE domain.
>
> ---
>
> ## W4&Q4: Evaluation is limited to MLE-Bench; generalizability remains questionable.
> While we agree that AI research encompasses tasks beyond MLE, we contend that MLE is the experimental backbone of modern AI research. To clarify the paper's focus, we will update the paper to specify that MARS targets the MLE aspects of the AI research cycle and will discuss extending the framework to broader AI research domains as future work.
>
> ---
>
>
> ## W5: Economic feasibility requires further discussion.
> To address economic feasibility, we conduct a new experiment demonstrating that MARS achieves superior performance at a cost of only $9.6 per task, which is less than both AIDE and AIRA-dojo. Please refer to our response to Reviewer ZXjX (for W3&Q1) for the details of these results.
>
> ---
>
> ## Q3: How ​​is the quality of the lessons evaluated?
> To measure the reliability and effectiveness of our lesson pool, we conducted a quantitative impact analysis: trajectory analysis reveals that while 36% of referenced lessons directly triggered a validation metric improvement, only 14% led to a decline, 9% led to the same score and 41% nodes required debugging. In the stochastic and high-noise environment of MLE, this represents a strong high-signal heuristic pool.
> Regarding how the accuracy or reliability of these lessons is measured, please refer to our response to Reviewer ZXjX (for Q3).
>
> ---
>
> ## Q5: Does the penalty term w in the Budget-Aware MCTS require tuning for different tasks?
>
> Our additional evaluations across Lite, Medium, and High splits (detailed in https://tinyurl.com/5t4msvdw - see Figure 7-9) demonstrate that w=−0.07 is a robust, general-purpose setting that does not require task-specific tuning. Across all task complexities, $w = -0.07$ consistently outperforms both $w=0$ (which doesn’t penalize latency) and $w=-0.15$ (which over-penalizes latency). These results confirm that $w = -0.07$ serves as a stable, default heuristic for resource-aware planning in long-horizon MLE tasks.
>
> ---
>
> ## L1: Add a "Limitations" subsection.
>
> We will include a "Limitations" section in the revised manuscript to address the mentioned limitations.

---

> > ### Author Rebuttal · Reviewer_JnZ4 · 2026-04-03
> >
> > Thank you for the detailed rebuttal and the additional experiments. The new ablation comparing Comparative Reflective Memory against the Empirical Analysis fallback was very helpful. The distinctions from ExpeL are also now clearer, though I encourage the authors to make these explicit in the revised related work section rather than leaving them only in the rebuttal.

---

> > > ### Author Response · Authors · 2026-04-04
> > >
> > > Thank you for the positive feedback and for increasing your score. We are glad that the additional ablation study and clarifications regarding ExpeL addressed your concerns. We will certainly incorporate the detailed distinctions between our work and ExpeL into the Related Work section of the final manuscript to ensure they are explicit for the readers.

---

### Decision · Program_Chairs · 2026-04-30

**Decision:**

Accept (regular)

**Comment:**

This paper proposes a system for improving automated AI research, through combining search (MCTS specifically, with an added criteria representing the experimental cost) with code generation pipelines, which is evaluated on MLE-Bench. All reviewers were positive - some noted that while the individual components were not necessarily novel, the system integration was well done and the empirical results convincing. All reviewers recommended acceptance, hence I recommend accept.